# PhysiX: A Foundation Model for 2D Physics Simulations

## Abstract

While foundation models have achieved remarkable success in domains like video, image, and language by scaling on massive datasets, this progress has not yet translated to physics simulation. A primary bottleneck is data scarcity: while millions of images, videos, and textual resources are readily available on the internet, the largest physics simulation datasets contain only tens of thousands of samples. This data limitation makes large models prone to overfitting and has confined physics applications to small models, which struggle with complex domains and long-range predictions. Furthermore, the drastic variations in scale and structure across physics datasets—a heterogeneity not typically found in vision or language—further amplify the challenges of scaling up multitask training. We introduce **PhysiX**, a family of large-scale foundation models for physics simulation. PhysiX is an autoregressive generative model composed of a discrete tokenizer, which converts heterogeneous physical processes to sequences of tokens, and a Transformer that models these sequences via next-token prediction. To mitigate the rounding error in the discretization process, PhysiX additionally incorporates a specialized refinement module. Extensive experiments on 2D datasets in The Well benchmark show superior performance of PhysiX over existing foundation models and strong task-specific baselines. Our results demonstrate that PhysiX benefits from synergistic learning through joint training on diverse simulation tasks and can successfully transfer knowledge from natural videos to the physical domain. We further analyze PhysiX's generalization to unseen domains and conduct careful ablation studies to validate the impact of each design component.

## 1 Introduction

Simulating physical systems governed by partial differential equations (PDEs) is a cornerstone of modern science and engineering. From modeling climate and fluid dynamics to understanding galaxy formation and biological morphogenesis, PDE-based simulations enable us to predict, control, and optimize complex natural phenomena (Eyring et al., 2016; Berger & LeVeque, 2024; Biegler et al., 2003; Mohammadi & Pironneau, 2004; Cranmer et al., 2020; Lemos et al., 2023). Traditionally, physics simulations have relied on numerical solvers that discretize and integrate governing equations over space and time. While highly accurate, such methods are computationally intensive, often requiring specialized hardware and expert-tuned software (Goldberg et al., 2022). This high cost has led to growing interest in machine learning (ML)-based surrogates, which aim to approximate simulation outputs at a fraction of the expense (Sun et al., 2020; Tao & Sun, 2019; Haghighat et al., 2021). Recent work has shown that deep neural networks can learn surrogate models for a range of PDE-driven systems, enabling orders-of-magnitude reductions in inference time (Torlai et al., 2018; Ryczko et al., 2019; Choudhary et al., 2022; Siahkoohi et al., 2023; Gopakumar et al., 2024).

Despite these advances, current ML-based surrogates remain largely task-specific. Most methods are designed for a single system and trained from scratch using individual datasets. These models typically struggle to adapt when simulation parameters such as domain geometry, boundary conditions, or physical constants change, and often require significant retraining or architectural modification to maintain accuracy (Franco et al., 2023; Zhang & Garikipati, 2023; Nguyen et al., 2024a; Gupta & Brandstetter, 2022). Moreover, since they are trained separately for each task, they fail to capture shared inductive biases across domains, such as spatiotemporal locality or conservation laws. To address similar limitations of task-specific models in other domains, researchers have increasingly

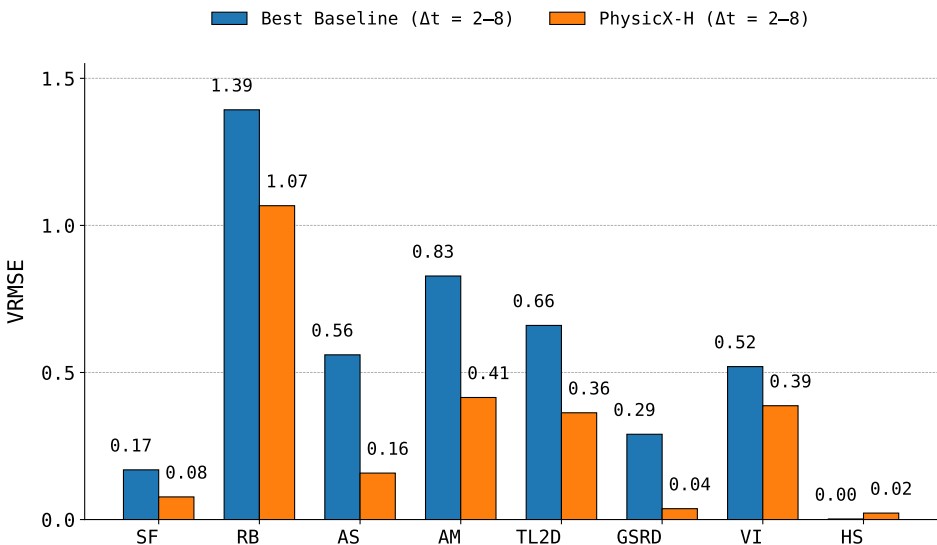

Figure 1: We propose **PhysiX**, a foundation model pretrained for physics simulations. We train PhysiX over a collection of 8 physics simulation tasks of the Well benchmark, resulting in a multi-task model that outperforms previous single-task baselines. We report VRMSE (lower is better) averaged across different physical properties and lead time between 9-26 frames for each task.

adopted the foundation model paradigm, where a large model is first pretrained on a large set of diverse data, before being finetuned for specific tasks (Brown et al., 2020; Bommasani et al., 2021).

The success of foundation models raises a natural question: *can we build a foundation model for physical simulations?* Unlike text or images, physics simulations pose unique challenges. First, simulation data is expensive to generate and inherently limited in volume. Even the largest public datasets contain only tens of thousands of spatiotemporal examples (Ohana et al., 2024), orders of magnitude smaller than the data used to train large language and vision models. In addition, physical systems exhibit substantial diversity in resolution, dimensionality, underlying equations, and physical domains: from turbulent fluids to elastic solids and chemical reaction-diffusion systems. Modeling such heterogeneity requires a flexible architecture and a training strategy capable of learning shared representations across domains while preserving task-specific fidelity. Together, these challenges make it non-trivial to scale the foundation model paradigm to physical simulations.

In this work, we introduce **PhysiX**, a novel foundation model for physical simulations. PhysiX comprises three main components: a universal discrete tokenizer, an autoregressive transformer, and a refinement module. We first train the tokenizer jointly on a diverse collection of physics datasets to compress continuous spatiotemporal fields at varying scales into sequences of discrete tokens, allowing the model to capture shared structural and dynamical patterns across domains. This shared discrete representation allows training on a unified token space analogous to language modeling. Building on this representation, we train a large-scale autoregressive transformer using a next-token prediction objective over the combined tokenized corpus. To improve generalization, we initialize both the tokenizer and the autoregressive model from pretrained checkpoints of high-capacity video generation models, enabling PhysiX to leverage strong spatiotemporal priors from natural videos. Finally, to address the quantization error in tokenization, PhysiX incorporates a lightweight refinement module that reconstructs fine-scale details from predicted token sequences.

Empirically, we find that PhysiX significantly outperforms task-specific baselines and foundation models on The Well benchmark (Ohana et al., 2024), demonstrating superior long-range prediction and better generalization across tasks. Figure 1 highlights these results. Our experiments show that PhysiX successfully extended the foundation model paradigm to physical simulations and demonstrated that joint training across multiple simulations enables synergistic learning. These results present compelling evidence that foundation models can serve as unified surrogates for diverse physical systems, bringing us closer to general-purpose and scalable tools for scientific computing.

## 2 RELATED WORKS

**Data-driven Physics Simulation** Traditional physics simulation relies on numerical methods, such as the finite element or finite volume methods, to solve governing differential equations. While effective, these approaches are often computationally expensive, limiting their scalability and applicability to high-resolution or long-term forecasting. Machine learning offers promising alternatives to accelerate or supplement traditional PDE solvers (Subramanian et al., 2023; Karniadakis et al., 2021), which have broadly diverged into two main paradigms. On one hand, physics-informed neural networks (PINNs) embed governing equations directly into the loss function, reducing the need for large datasets and ensuring physically plausible solutions (Raissi et al., 2019). However, this reliance on known physical laws makes PINNs unsuitable for systems where the underlying principles are not fully understood. On the other hand, purely data-driven surrogate models implicitly learn system dynamics from observed data (Lu et al., 2019). This area has evolved from early convolutional architectures like U-Net (Ronneberger et al., 2015; Zhu & Zabaras, 2018) to modern neural operators that learn mappings between infinite-dimensional function spaces (Kovachki et al., 2023; Lu et al., 2021). Prominent examples include Fourier Neural Operators (FNOs), which leverage Fast Fourier Transforms for global convolution (Li et al., 2021), Transformer-based models that use attention to capture long-range dependencies Li et al. (2022); Kissas et al. (2022), and Graph Neural Network (GNN) based operators that handle complex geometries by operating on unstructured meshes (Li et al., 2020; Brandstetter et al., 2022). While most of these methods focus on grid-based PDEs, TIE (Shao et al., 2022) introduces a specialized transformer architecture for particle-based dynamics in computer graphics. However, TIE focuses on learning dynamics within specific systems and does not address the challenge of generalization across multiple, heterogeneous scientific simulation tasks.

Despite these advancements, current data-driven simulators face critical limitations. Many models struggle with autoregressive, long-range predictions, where errors tend to accumulate over time (Lippe et al., 2023). Furthermore, most approaches are highly specialized: they are trained and optimized for a single physical system, a narrow range of parameters, or a specific set of governing equations. While these models can generalize within a given physical domain, they generally fail to transfer knowledge across distinct domains without substantial retraining or architectural modification.

**Foundation Models** The concept of foundation models first emerged in the context of transfer learning Zhuang et al. (2020), where a model trained on large-scale data in one domain can be easily fine-tuned to perform multiple tasks in adjacent domains. Notable early examples include self-supervised learning on ImageNet-1K, a dataset of natural images Chen et al. (2020); He et al. (2020); Oquab et al. (2023). These pre-trained vision models proved to be versatile for a wide range of downstream applications such as medical imaging Kim et al. (2022). More recently, several works have focused on building foundation models for domain-specific use cases such as remote sensing Reed et al. (2023), weather forecasting Nguyen et al. (2023), and material design Takeda et al. (2023).

Recent efforts toward physics foundation modeling include Multiple Physics Pretraining (MPP), which jointly trains transformers on heterogeneous spatiotemporal systems to improve transfer and generalization across physical domains McCabe et al. (2024), and DPOT (Denoising Operator Transformer), which injects noise into PDE trajectories and uses an autoregressive denoising objective to stabilize large-scale PDE pretraining Hao et al. (2024). MPP and DPOT are both transformer-based architectures trained to predict the next frame given historical context. MPP employs axial attention to model temporal and spatial dimensions separately, whereas DPOT combines a temporal aggregation layer with multiple Fourier attention layers. Concurrent with these works, other transformer-based approaches have been proposed. BCAT trains a causal autoregressive transformer directly on pixel-space sequences (Liu et al., 2025); unlike PhysiX, it relies on zero-padding to handle heterogeneity and lacks the computational efficiency of latent-space modeling for high-resolution data. Zebra (Serrano et al., 2025) utilizes discrete tokenization similar to our approach but focuses on in-context learning to adapt to varying parameters within a single system, rather than generalizing across entirely distinct physical domains. Fluid-LLM (Zhu et al., 2024) leverages pretrained LLM backbones for physics but restricts itself to one-step predictions via a graph decoder, distinguishing it from the multi-step autoregressive modeling of foundation models. Another significant development, Poseidon, introduces multiscale operator transformers with time-conditioned layers and semigroup-based training to scale foundation modeling for fluid dynamics Herde et al. (2024).

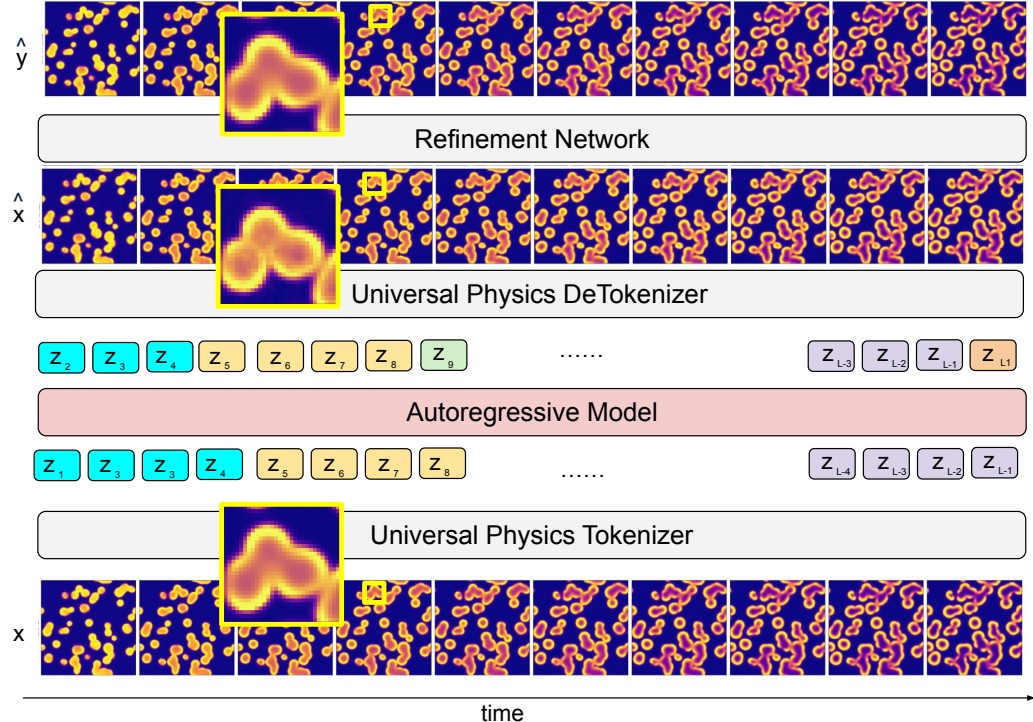

Figure 2: **The overall design of PhysiX**. PhysiX consists of a universal physics tokenizer, an autoregressive model, and a refinement network. Given input frames $x_1, \ldots, x_N$ from any simulation, the tokenizer discretizes each frame into a sequence of discrete tokens, where the $j$th token of frame $i$ is denoted as $\{z_i^j\}$. The autoregressive model then generates predictions in this discrete token space, which are converted back to pixel-level predictions $\hat{x}$ by the de-tokenizer. A refinement module is incorporated to mitigate artifacts caused by the discretization error, such as blocky, pixelated outputs (visualized in yellow boxes), and produce the final sharper and more detailed output $\hat{y}$.

## 3 METHOD

The PhysiX framework consists of three main components: a discrete tokenizer, an autoregressive model, and a refinement module. Given a physical simulation of $k$ frames, $x_{1:k} = (x_1, x_2, \ldots, x_k)$, where each frame $x_i \in \mathbb{R}^{H \times W \times C}$, the tokenizer converts the input data into a sequence of discrete tokens $z_1, z_2, ..z_L$ in a fixed vocabulary set $V$. We employ temporal and spatial compression such that the total sequence length is $\frac{H}{s_{HW}} \times \frac{W}{s_{HW}} \times \frac{T}{s_T}$, where $s_{HW}$ is the spatial compression ratio and $s_T$ is the temporal compression ratio. PhysiX then applies autoregressive modeling on this discrete sequence in a way similar to autoregressive large language models.

### 3.1 UNIVERSAL PHYSICS TOKENIZER

**Tokenizer architecture** Our tokenizer transforms an input video into a compact sequence of discrete tokens using a U-Net-style encoder-decoder architecture. The encoder first partitions the video frames into smaller patches and then processes them through a series of residual blocks interleaved with downsampling layers. This combination of patch embedding and downsampling blocks produces a sequence of latent frames with a predefined compression ratio. This latent representation is then quantized into discrete tokens using Finite-Scalar Quantization (FSQ) (Mentzer et al.). Symmetrically, the decoder reconstructs the original video from these tokens, mirroring the encoder's structure by replacing downsampling with upsampling blocks.

To support the downstream autoregressive model, the tokenizer must enforce temporal causality, ensuring that the encoding of any given video frame remains independent of all future frames. We achieve this using factorized spatio-temporal 3D convolutions, which combine a 2D spatial

convolution with a 1D temporal convolution using causal padding to prevent information leakage. For PhysiX, we train this tokenizer to achieve an $8\times$ spatial and $4\times$ temporal compression.

**Handling Data Heterogeneity**  Training a single, universal tokenizer across diverse simulation datasets presents a significant challenge due to data heterogeneity in channel dimensionality and physical semantics of the input fields. For instance, a shear flow simulation provides velocity and pressure fields, while a Gray-Scott reaction-diffusion dataset contains the concentrations of two chemical species. To address this, we introduce two modifications to the tokenizer architecture. First, we adjust the encoder's initial embedding layer to accept a fixed union of all input channels available across the datasets. Since each dataset contains only a subset of channels, for any given data point, we pad missing channels with dedicated, learnable 2D tensors. This enables the model to handle any subset of channels within a unified framework. Symmetrically, the decoder's final layer is adapted to predict the full union of all possible output fields. During training, however, the loss is computed only on the fields present in the ground-truth data for each specific sample. These modifications enable the model to train jointly on datasets with varying structures, facilitating the training of a shared embedding space that captures common physical principles and enables knowledge transfer across different simulation domains.

**Balancing Multitask Objectives**  Different physics simulation datasets vary significantly in size. If we were to sample uniformly from the combined data, larger datasets would dominate the training process, causing the model to under-learn the dynamics of less-represented domains. To prevent this bias, we employ a stratified sampling strategy that gives each task equal importance. Specifically, at each training iteration, we first sample a dataset with uniform probability and then select a random trajectory from within that chosen dataset. This two-stage process ensures that each physical domain contributes an equal number of sequences over time, which is crucial for developing a versatile and generalizable physics foundation model.

## 3.2 Autoregressive Generative Models

Following the tokenization stage, we train a large-scale autoregressive model to simulate physical dynamics within the discrete latent space. We employ a decoder-only Transformer architecture that is trained on a next-token prediction objective. Given sequences of discretized tokens $z_{1:L}$ representing physical simulations, the AR training objective can be formulated as:

$$\mathcal{L}_{\text{AR}} = -\sum_{i=1}^{L-1} \mathbb{E}_z \left[ \log p(z_{i+1}|z_{1:i}) \right], \tag{1}$$

where $L = \frac{HWT}{(s_{HW})^2(s_T)}$ is the total sequence length.

**3D RoPE for Varying Spatial Dimensions**  To effectively model the spatiotemporal nature of the simulation data, the autoregressive model incorporates 3D Rotary Position Embeddings (RoPE) (Su et al., 2024) to encode relative positional information of the token sequence. A key aspect of our implementation is its ability to handle variable spatial resolutions during training, a common challenge when working with diverse simulation datasets. Instead of resizing inputs or interpolating position embeddings, we dynamically adjust the positional encodings by truncating the 3D RoPE frequencies along the height and width dimensions to match the spatial dimensions of the current input. This approach, requiring minimal modification to the standard RoPE module, allows for the seamless handling of mixed-resolution data without compromising performance. We found this simple strategy performed on par with more complex interpolation techniques (Peng et al.; Zhuo et al.).

**Training and Inference**  During training, the model sees a sequence of 13 frames, which the tokenizer compresses into a sequence of $5 \times H/8 \times W/8$ latent tokens. The tokens corresponding to the first 5 frames serve as context, and we train the model with the next-token prediction loss on the tokens corresponding to the remaining 8 frames. During inference, PhysiX generates predictions token-by-token, analogous to an LLM. Specifically, given the latent tokens of the 5 context frames, the model autoregressively generates the tokens corresponding to the next 8 frames. To obtain longer-horizon rollouts, we use the last 5 generated frames as the new input context and repeat the process.

## 3.3 REFINEMENT MODULE

The refinement module is a convolutional neural network designed to correct artifacts introduced by the discrete tokenization process. As illustrated in an example in Figure 2, the coarse predictions from the autoregressive (AR) model, $\hat{x}$, exhibit a pattern that is similar to quantization noise in the center, whereas the ground truth data (bottom) $x$ is noise-free. Our refinement module successfully suppresses these artifacts to produce a clean output $\hat{y}$. These errors are inherent to the discrete tokenizer, which was originally developed for natural videos where minor imprecisions are often imperceptible. In the context of physical simulation, however, where high precision is critical, such noise can significantly degrade performance.

We train the refinement module as a post-processing step after the AR model has been fully trained. To generate training data, we first use the trained AR model to produce coarse predictions for the entire training set. The refinement module is then trained to map these coarse predictions to the corresponding ground-truth frames, operating entirely in pixel space to directly address visual artifacts. For the architecture, we adopt the ConvNeXt-U-Net model from the Well benchmark and optimize it using a Mean Squared Error (MSE) loss. While our architecture is similar to the benchmark's baseline, our learning objective differs: the module learns to correct the AR model's output rather than predicting the next frame from historical context. We train a separate refinement module for each dataset. Additional implementation details are available in Appendix C.

## 4 EXPERIMENTS

We conduct a comprehensive evaluation of PhysiX, comparing it against state-of-the-art methods to assess its performance and generalization capabilities. Additionally, we perform careful ablation studies to validate the impact of our key architectural and training design choices.

**Benchmark and Evaluation Metrics** We train and evaluate PhysiX across eight simulation tasks from the Well benchmark (Ohana et al., 2024), as shown in Tables 1 and 2, which collectively provide 1.67TB of data for training. We selected 8 representative datasets to balance diversity and feasibility. The complete Well corpus exceeds 10 TB, making full training computationally intensive. Our chosen subset retains coverage of core physical phenomena while ensuring reproducibility. Following the official protocol, we evaluate performance using the Variance-Weighted Root Mean Squared Error (VRMSE), averaged across all physical channels for each dataset. For the two tasks `helmholtz_staircase` and `acoustic_scattering (maze)`, we exclude static channels that remain constant over time from the evaluation. We consider both next-frame prediction and long-horizon rollout settings.

**Baselines** We compare PhysiX against MPP (McCabe et al., 2024) and DPOT (Hao et al., 2024), two existing foundation models for multi-physics PDEs. In addition, we include the four baselines provided by the Well benchmark: Fourier Neural Operator (FNO) (Li et al., 2021), Tucker-Factorized FNO (TFNO) (Kossaifi et al.), U-Net (Ronneberger et al., 2015), and U-Net with ConvNeXt blocks (C-U-Net) (Liu et al., 2022).

**Scaling** We train a family of PhysiX models at four scales: 250M (PhysiX-S), 700M (PhysiX-M), 2B (PhysiX-L), and 4B (PhysiX-H) parameters. For our largest model, PhysiX-H, we initialize the

Table 1: **Next-frame prediction performance across 8 datasets on the Well benchmark**. We report VRMSE (lower is better) averaged across different fields for each dataset.

| Dataset | The Well Baselines | | | | Foundation Models | | | Ours | |
|---|---|---|---|---|---|---|---|---|---|
| | FNO | TFNO | U-Net | C-U-Net | MPP | DPOT | Poseidon | PhysiX-S | PhysiX-H |
| shear_flow | 1.189 | 1.472 | 3.447 | 0.8080 | 0.2016 | 0.3120 | 0.9925 | 0.0740 | **0.0700** |
| rayleigh_benard | 0.8395 | 0.6566 | 1.4860 | 0.6699 | 0.3332 | 0.4014 | 0.9630 | 0.1840 | **0.1470** |
| acoustic_scattering (maze) | 0.5062 | 0.5057 | 0.0351 | **0.0153** | 0.1773 | 0.0824 | 0.8242 | 0.1400 | 0.0960 |
| active_matter | 0.3691 | 0.3598 | 0.2489 | 0.1034 | 0.1664 | 0.3040 | 0.3839 | 0.1100 | **0.0904** |
| turbulent_radiative_layer_2D | 0.5001 | 0.5016 | 0.2418 | **0.1956** | 0.3676 | 0.3490 | 0.3002 | 0.2299 | 0.2098 |
| viscoelastic_instability | 0.7212 | 0.7102 | 0.4185 | 0.2499 | 0.2729 | 0.2538 | 0.8362 | 0.3669 | **0.2370** |
| gray_scott_reaction_diffusion | 0.1365 | 0.3633 | 0.2252 | 0.1761 | 12.5853 | 0.2602 | 3.0494 | 0.0396 | **0.0210** |
| helmholtz_staircase | **0.00046** | 0.00346 | 0.01931 | 0.02758 | 0.0863 | 0.0745 | 0.2887 | 0.0146 | 0.0180 |
| Average Rank (↓) | 6.25 | 6.50 | 5.63 | 3.38 | 5.50 | 4.88 | 7.88 | 3.13 | **1.88** |

Table 2: **Long-horizon prediction performance across 8 datasets on the Well benchmark.** We report VRMSE (lower is better) averaged across different fields for each dataset. We report averaged results over different ranges of lead time: 2-8, 9-26 and 27-56 frames.

| Dataset | $\Delta t = 2{:}8$ | | $\Delta t = 9{:}26$ | | $\Delta t = 27{:}56$ | |
|---|---|---|---|---|---|---|
| | Baseline | PhysiX-H | Baseline | PhysiX-H | Baseline | PhysiX-H |
| shear_flow | 0.169 | **0.077** | 0.687 | **0.153** | 1.340 | **0.236** |
| rayleigh_benard | 1.393 | **1.067** | 0.963 | **0.741** | 1.026 | **0.847** |
| acoustic_scattering (maze) | 0.560 | **0.158** | **0.920** | 1.246 | **1.341** | 2.189 |
| active_matter | 0.828 | **0.415** | 1.446 | **0.974** | 1.635 | **1.320** |
| turbulent_radiative_layer_2D | 0.660 | **0.363** | 1.040 | **0.693** | 1.331 | **0.953** |
| gray_scott_reaction_diffusion | 0.290 | **0.037** | 0.712 | **0.390** | **0.615** | 0.894 |
| viscoelastic_instability | 0.520 | **0.387** | — | — | — | — |
| helmholtz_staircase | **0.002** | 0.022 | **0.003** | 0.071 | — | — |

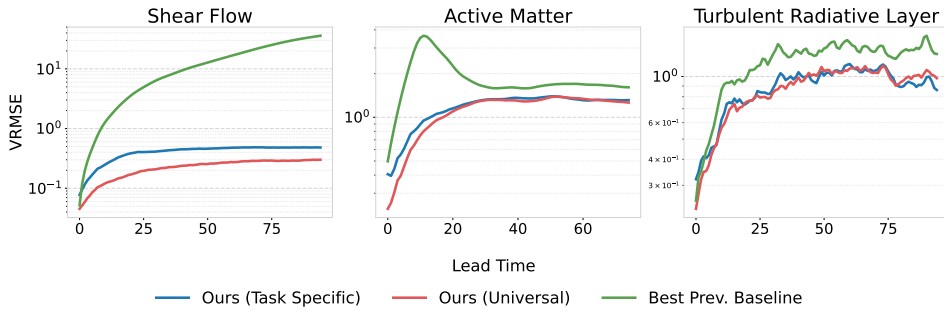

Figure 3: **Long-horizon prediction performance.** We show VRMSE (lower is better) of the best baseline vs. our method across different lead times on three simulation datasets.

weights from a pre-trained video generative model of an equivalent size (Agarwal et al., 2025). This transfer learning from natural videos significantly improves performance and generalization of large models, which we discuss in more detail in Section 4.4. In our main comparison tables, we report results primarily for the PhysiX-S and PhysiX-H variants due to space constraints.

### 4.1 NEXT-FRAME PREDICTION

As shown in Table 1, our largest model, PhysiX-H, achieves state-of-the-art performance on 5 of the 8 datasets, achieving the top average rank of 1.88. Our smallest model, PhysiX-S, also performs strongly, ranking second overall with an average rank of 3.13 and outperforming the best specialist baseline, C-U-Net (3.38). The performance gains are particularly notable on datasets like shear_flow and rayleigh_benard, where PhysiX-H reduces the VRMSE by 63% and 56% respectively, compared to the strongest baseline. The only task where PhysiX is inferior to the baselines is helmholtz_staircase. We found that this simulation exhibits a strong quasi-periodic pattern, which is an ideal scenario for Fourier-based architectures like FNO and TFNO, explaining their superior performance on this specific task.

When compared to other foundation models, PhysiX demonstrates superior performance and parameter efficiency. Even our smallest 250M parameter model, PhysiX-S, significantly outperforms the larger MPP (400M) and DPOT (500M) models. We attribute this advantage to two key architectural choices. First, we use a robust method for handling heterogeneous fields by treating each physical channel separately across all datasets. In contrast, DPOT's strategy of padding to the maximum number of channels can cause ambiguity for the model, since different physical quantities (e.g., 'pressure' from one dataset and 'velocity' from another) are forced to share the same embedding weights. Second, PhysiX employs a unified autoregressive Transformer that processes space and time jointly in the latent space. Unlike MPP and DPOT which interleave space and time modeling, our architecture learns their intricate coupling directly, allowing it to capture more complex physical dynamics. Previous work in other physical domains such as weather and climate (Nguyen et al., 2023; 2024b; Bodnar et al., 2025) has also shown strong evidence of the superiority of simple and scalable architectures over more specialized methods.

Table 3: **Comparison of multi- and single-task models.** We report next-frame and long-horizon prediction results on the Well benchmark for the multi-task and single-task models.

| Dataset | $\Delta t = 1$ | | $\Delta t = 2:8$ | | $\Delta t = 9:26$ | | $\Delta t = 27:56$ | |
|---|---|---|---|---|---|---|---|---|
| | Spec. | Univ. | Spec. | Univ. | Spec. | Univ. | Spec. | Univ. |
| shear_flow | **0.0689** | 0.070 | 0.236 | **0.077** | 0.378 | **0.153** | 0.452 | **0.236** |
| rayleigh_benard | **0.137** | 0.147 | **0.436** | 1.067 | **0.522** | 0.741 | 0.848 | **0.847** |
| turbulent_radiative_layer | 0.359 | **0.201** | 0.565 | **0.363** | 0.792 | **0.693** | 1.014 | **0.953** |
| active_matter | 0.150 | **0.090** | 0.844 | **0.415** | 1.177 | **0.974** | 1.352 | **1.320** |
| gray_scott_reaction | 0.0418 | **0.0210** | 1.487 | **0.037** | 15.965 | **0.390** | 62.484 | **0.894** |
| viscoelastic_instability | 0.251 | **0.237** | 0.764 | **0.387** | — | — | — | — |

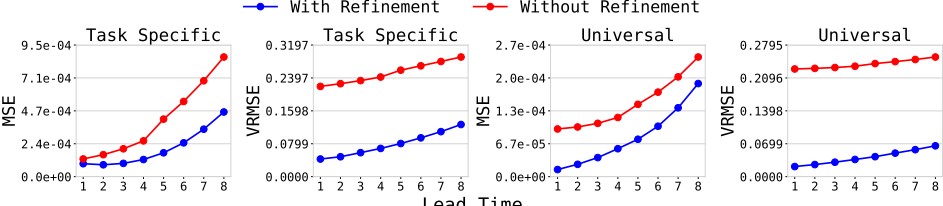

Figure 4: **Effect of refinement module.** We apply refinement module to both the multi-task and single-task AR model and study its effect on prediction errors. We report VRMSE and MSE (lower is better) over prediction windows ranging from 1 frame to 8 frames on the `gray_scott_reaction_diffusion` dataset.

## 4.2 LONG-HORIZON PREDICTION

While PhysiX performs competitively in next-frame prediction, its true strength lies in long-horizon simulation. As shown in Table 2, PhysiX-H achieves state-of-the-art performance on $18/21$ evaluation points across different forecasting windows, when compared to the best baseline *for each specific task*. The improvements are not only consistent but also significant in various tasks. For example, on `shear_flow`, PhysiX-H reduces VRMSE by over $97\%$ at the 6:12 horizon compared to the best-performing baseline (from 2.33 to 0.077). On `rayleigh_benard`, PhysiX-H achieves more than $90\%$ lower error across all rollout windows. Similar results are observed in `active_matter`, where PhysiX-H consistently achieves better performance at every forecast horizon, underscoring its robustness and adaptability across domains.

Figure 3 further illustrates the long-term behavior of PhysiX compared to the best baseline model in each dataset. In the early stages of the rollout, both models exhibit similar performance. However, as the lead time increases, the performance of the baseline models degrades rapidly due to compounding prediction errors. In contrast, PhysiX maintains low VRMSE across time steps, demonstrating much greater stability. This stability stems from the autoregressive nature of PhysiX, which allows the model to learn from full sequences of simulations rather than focusing solely on short-term prediction. This enables it to maintain stability and accuracy over extended rollouts, making it particularly well-suited for challenging multi-step prediction tasks. We also note that we use lead time as the basis for comparison instead of using a fixed window of frames as in the original The Well paper. This ensures a fairer comparison since PhysiX, DPOT, and MPP use $5$ frames as historical context, while the other baselines use $4$.

## 4.3 ABLATION STUDIES

We conduct a series of ablation experiments to study the effectiveness of our design. Specifically, this section compares the universal (multi-task) models and single-task models, studies the effectiveness of the refinement module, scaling properties, and generalization of PhysiX to unseen physical domains.

**General Model vs Task Specific Models** We compare the performance of the multi-task and single-task models on both one-frame and long-horizon prediction tasks. For the task-specific model, we followed the same setup as the universal model, including the model size, model architecture,

Table 4: **Prediction errors of different model sizes at various time horizons.** We report next-frame and long-horizon prediction errors for different model sizes across different datasets, highlighting the best (lowest) error in each horizon.

| Dataset | $\Delta t = 1$ | | | | | $\Delta t = 2{:}8$ | | | | | $\Delta t = 9{:}26$ | | | | |
|---|---|---|---|---|---|---|---|---|---|---|---|---|---|---|---|
| | 4B* | 4B | 2B | 700M | 250M | 4B* | 4B | 2B | 700M | 250M | 4B* | 4B | 2B | 700M | 250M |
| shear_flow | **0.070** | 0.071 | 0.075 | 0.073 | 0.074 | **0.077** | 0.094 | 0.112 | 0.096 | 0.115 | **0.153** | 0.198 | 0.216 | 0.166 | 0.221 |
| rayleigh_benard | **0.147** | 0.174 | 0.181 | 0.194 | 0.184 | **1.067** | 1.10 | 1.201 | 1.113 | 1.177 | **0.741** | 0.761 | 0.855 | 0.827 | 0.872 |
| acoustic_scattering (maze) | **0.096** | 0.106 | 0.110 | 0.120 | 0.140 | **0.158** | 0.200 | 0.211 | 0.237 | 0.282 | **1.246** | 1.270 | 1.284 | 1.242 | 1.257 |
| turbulent_radiative_layer | **0.210** | 0.368 | 0.421 | 0.312 | 0.350 | **0.363** | 0.427 | 0.443 | 0.450 | 0.886 | **0.693** | 0.714 | 0.758 | 0.730 | 1.163 |
| active_matter | **0.090** | 0.130 | 0.102 | 0.105 | 0.110 | **0.415** | 0.579 | 0.592 | 0.623 | 0.629 | **0.974** | 1.544 | 1.626 | 1.394 | 1.495 |
| gray_scott_reaction | **0.210** | 0.228 | 0.230 | 0.231 | 0.244 | 0.534 | 0.577 | **0.509** | 0.526 | 1.643 | 1.984 | 1.544 | 1.126 | **1.051** | 9.029 |
| viscoelastic_instability | **0.237** | 0.255 | 0.319 | 0.246 | 0.367 | **0.387** | 0.490 | 0.494 | 0.590 | 0.916 | — | — | — | — | — |
| helmholtz_staircase | 0.018 | 0.015 | 0.015 | **0.014** | 0.015 | 0.022 | 0.0224 | 0.019 | **0.017** | 0.018 | 0.071 | 0.0718 | **0.056** | 0.061 | 0.063 |

and training hyperparameters. The only difference is the training data. We report VRMSE across 8 datasets and different lead times in Table 3. Experiment results show that the universal model outperforms task-specific models, achieving lower VRMSE on the majority of datasets across different lead times. Our results show that joint multi-task training improves the performance of individual tasks, as the model can learn shared patterns across different physical processes.

**Effectiveness of Refinement Module** Figure 4 compares PhysiX with and without the refinement module at different prediction windows, for both the multi-task and single-task AR models. The refinement module reduces MSE and VRMSE metrics for both models on all prediction windows of the gray_scott_reaction_diffusion dataset, highlighting the effectiveness of the proposed refinement process. Most notably, with the help of the refinement model, the 8-frame prediction error (0.07) of our multi-task model, measured by VRMSE, is lower than the 1-frame prediction error of the best performing baseline on the Well benchmark (0.14).

### 4.4 SCALING ANALYSIS

We analyze the scaling properties of PhysiX by comparing four model sizes: 250M, 700M, 2B, and 4B. For the 4B model, we evaluate two versions, one trained from scratch and the other initialized from a pre-trained video model (4B*). For simplicity, we exclude the refinement module for all models in this ablation. As shown in Table 4, the model exhibits positive scaling from 250M to 700M parameters, with the 700M model consistently outperforming its smaller counterpart. However, this trend does not continue when scaling further from scratch; models with 2B and 4B parameters show no clear pattern of improvement over smaller models. We find that while these larger models achieve a lower training loss, their increased capacity leads to overfitting on the smaller physics datasets, resulting in poorer test performance.

PhysiX overcomes this scaling limitation by knowledge transfer from natural videos. The 4B* model, initialized from a video model checkpoint, performs the best on $7/8$ simulation tasks. This result indicates that PhysiX can successfully transfer strong spatiotemporal priors from large-scale video models to the physics domain. This knowledge transfer boosts performance and is crucial for enabling the model to scale to a much larger capacity than is feasible when training from scratch.

### 4.5 ADAPTATION TO UNSEEN SIMULATIONS

We evaluate the adaptability of PhysiX on two unseen simulations: euler_multi_quadrants (periodic b.c.) and acoustic_scattering (discontinuous). These tasks involve novel input channels and physical dynamics not encountered during training. To handle this distribution shift, we fully finetune the tokenizer for each task. We consider two variants of the AR model: PhysiX$_f$, which finetunes the pretrained model, and PhysiX$_s$, which trains from scratch using the Cosmos checkpoint as initialization. Further finetuning details are provided in Appendix C.

Table 5 shows that PhysiX$_f$ achieves the best performance on nearly all tasks and prediction horizons, only losing to C-U-Net on one-step prediction for one task, and the performance gap widens significantly as the horizon increases. Notably, PhysiX$_f$ consistently outperforms PhysiX$_s$ across all settings, highlighting its ability to effectively transfer knowledge to previously unseen simulations.

---

[1]The model predicts exactly identical tokens for these setup, hence the performance is exactly the same.

Table 5: **Performance on two simulation tasks unseen during training.** We compare both the finetuning version (PhysiX$_f$) and the scratch version (PhysiX$_s$) with the baselines.

| Models | euler_multi_quadrants (periodic b.c.) | | | | acoustic_scattering (discontinuous) | | | |
|--------|------------|------------|-------------|--------------|------------|------------|-------------|--------------|
| | $\Delta t = 1$ | $\Delta t =2{:}8$ | $\Delta t =9{:}26$ | $\Delta t =27{:}56$ | $\Delta t = 1$ | $\Delta t =2{:}8$ | $\Delta t =9{:}26$ | $\Delta t =27{:}56$ |
| PhysiX$_f$ | **0.105**$*^1$ | **0.188**$*^1$ | **0.358** | **0.642** | 0.038 | **0.057** | **0.443** | **1.168** |
| PhysiX$_s$ | **0.105**$*^1$ | **0.188**$*^1$ | 0.366 | 0.658 | 0.039 | 0.062 | 0.455 | 1.192 |
| FNO | 0.408 | 1.130 | 1.370 | – | 0.127 | 2.146 | 2.752 | 3.135 |
| TFNO | 0.416 | 1.230 | 1.520 | – | 0.130 | 2.963 | 3.713 | 4.081 |
| U-Net | 0.183 | 1.020 | 1.630 | – | 0.045 | 2.855 | 6.259 | 8.074 |
| C-U-Net | 0.153 | 4.980 | >10 | – | **0.006** | 5.160 | >10 | >10 |

## 5 CONCLUSION

PhysiX introduces a unified foundational model for general-purpose physical simulation across diverse systems. PhysiX's joint training approach enabled it to capture shared spatiotemporal patterns and adapt to varying resolutions and physical semantics. We show a single universally trained model significantly outperforms task-specific baselines on many domains. PhysiX demonstrates particularly strong performance on long-horizon rollouts, while maintaining stability and accuracy. The success of PhysiX highlights the potential of foundation models in accelerating scientific discovery .

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

## A    LIMITATIONS

Despite the promising success of PhysiX, we acknowledge that it has several key limitations.

**Generalization**. Existing foundation models typically have zero-shot generalization capabilities. For example, CLIP Radford et al. (2021), which was pretrained on a large set of vision-language data, can perform zero-shot classification on images for domain-specific applications. While PhysiX is trained on multiple datasets, generalizing to novel physical processes requires fine-tuning, as they may have unseen input channels or represent a drastically different dynamic system from those seen during training. We leave this to future work.

**Discretization Error**. The tokenization process introduces quantization errors, and while the refinement module helps mitigate this, residual errors can still affect the precision of long-term simulations. This is especially significant for datasets with low spatial or temporal variance which are much more sensitive to small perturbations. Exploring alternative tokenization schemes or end-to-end training of the tokenizer and autoregressive model could help minimize this error.

**Data Diversity**. PhysiX was only trained on 2D datasets, due to the architecture of the video tokenizer. This limits its direct applicability to 3D physical systems or systems with significantly different spatial structures. Future work could explore more flexible tokenization architectures that enable the compression of higher spatial dimensions, and include data from outside The Well.

## B    ARCHITECTURAL DETAILS

### B.1    UNIVERSAL TOKENIZER

The tokenizer processes video data with a compression factor of $4\times$ in time and $8\times$ in each spatial dimension. The architecture consists of three modules: a variable-aware patcher, a factorized 3D encoder-decoder, and a Finite Scalar Quantization (FSQ) bottleneck.

**Variable Padding and Patching**  To handle the heterogeneous input fields in different datasets, input frames with dynamic channel subsets are mapped to a fixed $C_{total} = 46$ channel tensor. Missing variables are padded with learnable tensors to ensure a consistent input shape for the encoder. This padded volume is then patchified with a patch size of $p = 4$ across space and time.

**Encoder-Decoder Architecture**  The backbone is a symmetric U-Net composed of 3D Resnet blocks, which utilize $(1, 3, 3)$ spatial and $(3, 1, 1)$ temporal causal convolutions to strictly preserve temporal causality. The network is configured with a base width of $C = 128$ channels. The encoder contains $L = 3$ resolution levels with channel multipliers of $[2, 4, 4]$, resulting in feature dimensions of $256, 512$, and $512$ respectively. Each level consists of $N = 2$ residual blocks.

Downsampling is performed via factorized strided convolutions. Given the patch size of $4$ across space and time, the encoder applies 1 additional spatial downsampling operation to achieve the target compression.

**Quantization.** The bottleneck projects the latent features to a dimension of $d = 16$ per token. We utilize Finite Scalar Quantization (FSQ) with $d_{embed} = 6$ and quantization levels set to $[8, 8, 8, 5, 5, 5]$, yielding a codebook size of $64,000$.

### B.2    AUTOREGRESSIVE TRANSFORMER

Our autoregressive transformer is a standard decoder-only architecture, incorporating 3D Rotary Position Embeddings (RoPE) to account for spatiotemporal data. We train models of varying sizes, ranging from 250M to 4B parameters. The specific architectural hyperparameters for each model configuration are detailed in Table 6.

Table 6: Hyperparameter configurations for the Autoregressive Transformer models.

| Model Size | Layers | Hidden Size | FFN Size | Attn. Heads |
|---|---|---|---|---|
| 250M | 8 | 1,024 | 4,096 | 8 |
| 600M | 8 | 2,048 | 8,192 | 16 |
| 2B | 12 | 3,072 | 12,288 | 24 |
| 4B | 16 | 4,096 | 14,336 | 32 |

### B.3 REFINEMENT MODULE

We used the ConvNeXt U-Net architecture with the same hyperparameters reported in Ohana et al. (2024) for the refinement module.

## C EXPERIMENTAL SETTINGS

**Data Normalization** For each input channel in each dataset, we computed the global mean and standard deviation across the spatial domain and all time steps in the training set. We used these task- and channel-specific statistics to normalize inputs during training and denormalize predictions for evaluation.

**Tokenizer** We trained the universal tokenizer on the 8 datasets in Table 1 for 1000 epochs with an effective batch size of 32, using the standard Mean Squared Error (MSE) loss without any special weighting. We optimize the models using AdamW (Kingma & Ba, 2014) with a base learning rate of $1e-3$, using a 10-epoch linear warmup, followed by a cosine decay schedule for the remaining 990 epochs. For model selection, we average the validation loss across all datasets after each training epoch and use the model with the lowest validation loss as the final tokenizer checkpoint.

**AR Model** For the autoregressive (AR) model, we trained for 10000 steps with an effective batch size of 32. We used Adam as the optimizer with a learning rate schedule similar to the tokenizer, where the number of warmup steps is set to 1000. We validated the model after every 100 training steps and used the best checkpoint for testing. For both tokenizer and AR training, we upsampled the smaller datasets to match the size of the largest one, ensuring the model learns from each dataset uniformly.

**Refinement Module** For each trajectory in the raw training data, we randomly sample a starting timestamp and run autoregressive generation to obtain the training data for the refinement module. We adopted MSE loss. We use a global batch size of 64 frames, a learning rate of $5e-3$ and a cosine decay learning rate scheduler. We trained each refinement model for 500 epochs on its respective data. Unlike the base model, which is trained in bfloat16 precision, we observe that using float32 precision is crucial to achieve high-quality outputs, especially for datasets with low spatial variance.

**Evaluation** After training, we tested the model on the held-out test set provided by the Well (Ohana et al., 2024). For the one-step setting, we evaluated the model on random sliding windows sampled from the test simulations. For the long-horizon setting, we always initiated the model from the beginning of each simulation. This adheres to the standard practice in the Well.

**Finetuning** To adapt PhysiX to an unseen task, we finetune both the tokenizer and the autoregressive model. Specifically, we finetune the tokenizer for 100 epochs and the autoregressive model for 1000 iterations, with similar learning rates and schedulers to pretraining. This means the compute requirement for each finetuning task is about 10% of that of pretraining. Section 4.5 shows that PhysiX was able to achieve strong performance even with this limited compute, demonstrating its usefulness for the broad research community.

## D ADDITIONAL EXPERIMENTS

## D.1 A FAIRER COMPARISON BETWEEN SCRATCH AND FINETUNED PHYSIX

Our initial comparison in Table 5 was unfair to the scratch PhysiX baseline. In this table, both PhysiX$_f$ (finetuned) and PhysiX$_s$ (scratch) used the same VAE tokenizer that was pretrained on 8 datasets. This gave the scratch model an unfair advantage by leveraging a pretrained component.

To rectify this, we performed a rigorous re-evaluation where we trained new VAE tokenizers from scratch for the unseen tasks (`euler_multi_quadrants (periodic b.c.)` and `acoustic_scattering (discontinuous)`) and then trained the transformer on top. The results in Table 7 demonstrate that under this fairer comparison, PhysiX$_f$ significantly outperforms the scratch baseline, confirming the strong benefit of pretraining.

Table 7: Comparison of the finetuning version (PhysiX$_f$) and the scratch version (PhysiX$_s$) on two simulation tasks. The scratch baseline uses scratch-trained VAE tokenizers.

| Models | euler_multi_quadrants (periodic b.c.) | | | | acoustic_scattering (discontinuous) | | | |
|---|---|---|---|---|---|---|---|---|
| | $\Delta t = 1$ | $\Delta t = 2:8$ | $\Delta t = 9:26$ | $\Delta t = 27:56$ | $\Delta t = 1$ | $\Delta t = 2:8$ | $\Delta t = 9:26$ | $\Delta t = 27:56$ |
| PhysiX$_f$ | **0.105** | **0.188** | **0.358** | **0.642** | **0.038** | **0.057** | **0.443** | **1.168** |
| PhysiX$_s$ | 0.205 | 0.303 | 0.464 | 0.754 | 0.078 | 0.114 | 0.689 | 1.554 |

## D.2 EFFECTS OF THE REFINEMENT MODULE

Table 8 compares the performance of PhysiX with and without the refinement module. We observed that the refinement module primarily contributes to performance on two tasks: `gray_scott_reaction_diffusion` and `turbulent_radiative_layer_2D`. For example, in `gray_scott_reaction_diffusion`, the tokenizer introduced "blocky" artifacts that degraded VRMSE, and the refinement module effectively smoothed these out (see `https://imgur.com/a/iTiKoh3` for visualization).

Table 8: Performance of PhysiX with and without the refinement module.

| Task | PhysiX w/ refinement | PhysiX w/o refinement |
|---|---|---|
| `shear_flow` | 0.0700 | 0.0713 |
| `rayleigh_benard` | 0.1470 | 0.1477 |
| `acoustic_scattering (maze)` | 0.0960 | 0.0984 |
| `turbulent_radiative_layer_2D` | 0.2098 | 0.3430 |
| `active_matter` | 0.0904 | 0.0913 |
| `gray_scott_reaction_diffusion` | 0.0210 | 0.2290 |
| `viscoelastic_instability` | 0.2370 | 0.2397 |
| `helmholtz_staircase` | 0.0180 | 0.0181 |

## D.3 LONGER-HORIZON RESULTS

Different tasks in The Well have different numbers of time steps. Among the tasks we considered in this paper, only `gray_scott_reaction_diffusion` (1000), `rayleigh_benard` (200), and `shear_flow` (200) have more than 100 steps. In addition to the long-horizon results in Table 2, we report the performance of PhysiX and the best baseline up to 100 steps ahead in Table 9. Similar to the results in the main text, PhysiX significantly outperforms the baseline in 5/6 tasks, demonstrating the strong stability of our proposed method.

Table 9: Performance of PhysiX and the best baseline with up to 100 prediction steps.

| Dataset | $\Delta t = 57 : 76$ | | $\Delta t = 77 : 95$ | |
|---|---|---|---|---|
| | PhysiX | Baseline | PhysiX | Baseline |
| `shear_flow` | **0.455** | 20.482 | **0.463** | 20.482 |
| `rayleigh_benard` | **0.897** | 1.236 | **0.963** | 1.017 |
| `acoustic_scattering` (maze) | 2.798 | **1.774** | 3.001 | **2.750** |
| `active_matter` | **1.309** | 1.665 | - | - |
| `turbulent_radiative_layer_2D` | **1.073** | 1.379 | **0.988** | 1.340 |
| `gray_scott_reaction_diffusion` | **0.964** | 23.072 | **1.341** | 32.048 |

### D.4 TRAINING AND INFERENCE EFFICIENCY OF PHYSIX

**Training** We trained PhysiX, MPP, DPOT, and Poisedon on the same $8\times$ A100 GPUs for 10000 iterations with an effective batch size of 32. The smallest PhysiX-S took 3 hours to finish, while the largest model PhysiX-H took 19.5 hours. MPP, DPOT, and Poisedon took 14 hours, 14 hours, and 12 hours to finish training.

**Inference** Table 10 shows the inference efficiency of PhysiX and the baseline methods in terms of throughput, peak memory usage, and GFLOPs, measured on one-step prediction for the `active_matter` task.

Table 10: Inference efficiency of PhysiX and the baselines.

| Method | Speed (frames/s) | Peak memory usage (GBs) | GFLOPs |
|---|---|---|---|
| C-U-Net | 96.0 | 0.2 | 20.75 |
| MPP | 9.4 | 2.2 | 545.57 |
| DPOT | 26.0 | 2.3 | 535.52 |
| Poseidon | 54.7 | 2.5 | 143.69 |
| PhysiX-S | 3.5 | 4.1 | 818.19 |
| PhysiX-H | 0.65 | 11.2 | 15367.39 |

While PhysiX has lower inference throughput due to the autoregressive generation, this is an expected trade-off for the significant performance gains observed (lower VRMSE and better stability). Furthermore, even at 0.65 frames/s of the largest model, PhysiX remains orders of magnitude faster than traditional numerical solvers.

## E COMPUTE RESOURCES

We trained the tokenizer and PhysiX on $8\times$ 40GB A100 devices, and evaluated using $1\times$ 40GB A100 device for each task. We trained PhysiX for 24 hours on $8\times$A100s for 8 datasets. This is approximately equal to the combined cost of training the best baseline model for each dataset at current market rate cloud compute costs [1]. Each model in The Well required 12 hours on $1\times$H100 (Ohana et al., 2024), for a total time of 96 H100 hours when only considering the best model for each dataset, or about half the A100 hours used by PhysiX.

## F LICENSES

Cosmos Agarwal et al. (2025) is licensed under Apache-2.0, and the Well Ohana et al. (2024) benchmark follows BSD-3-Clause license. We respect the intended use of each artifact and complied with all license requirements.

---

[1]Using pricing from Lambda Labs

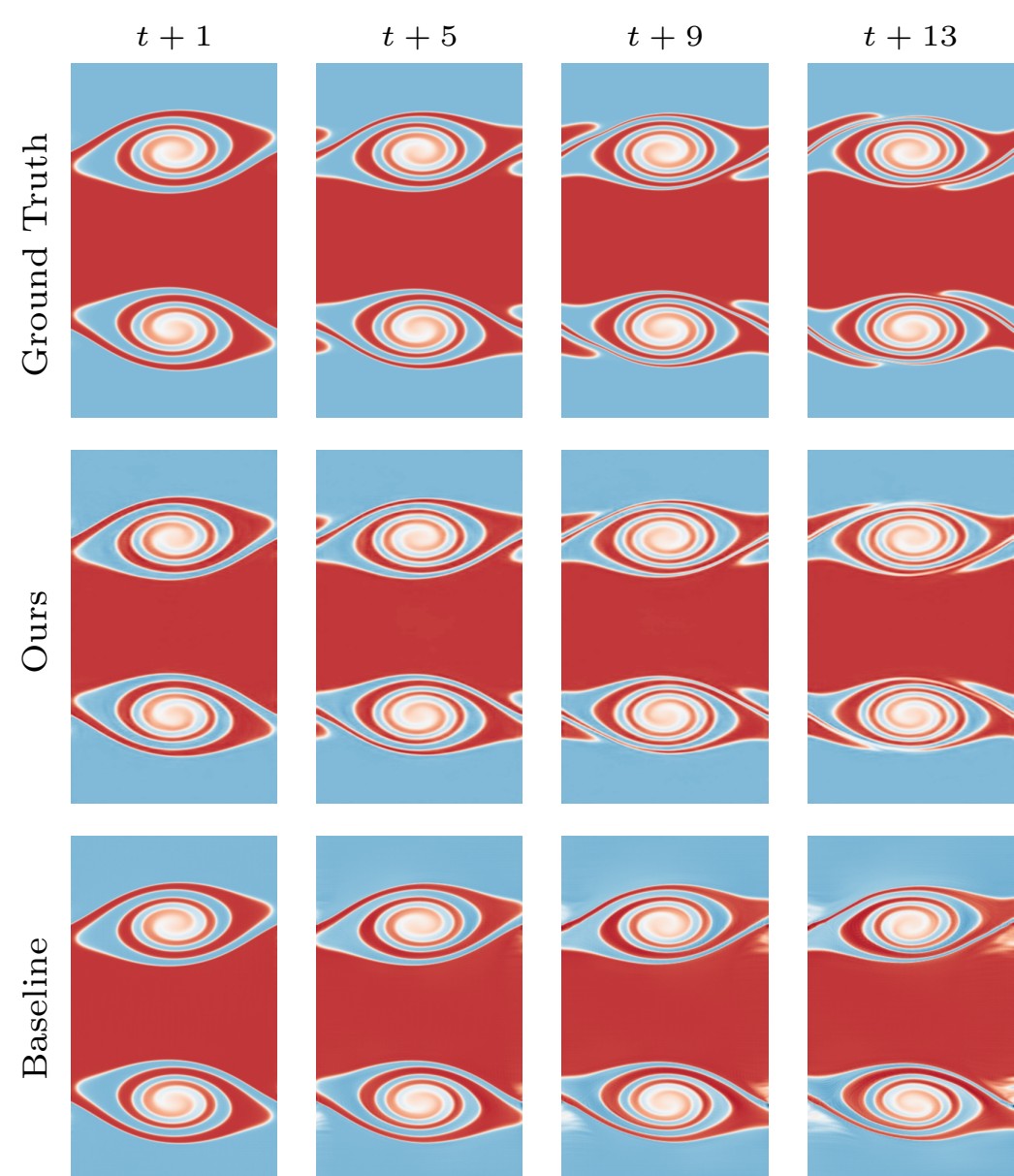

Figure 5: **Qualitative Comparisons on `shear_flow` Dataset.** We compare the prediction of PhysiX with the ground truth and the prediction of the best baseline model at lead times of 1,5,9,13 frames.

## G    QUALITATIVE RESULTS

We provide additional visualizations of the PhysiX's prediction results on `shear_flow` (Figure 5), `viscoelastic_instability` (Figure 6), `rayleigh_benard` (Figure 7) and `gray_scott_reaction_diffusion` (Figure 8). We compare the prediction of PhysiX with the ground truth and the prediction of baseline models at various lead times. PhysiX shows consistent improvement over baselines across all lead times. The improvements on longer lead times are more pronounced.

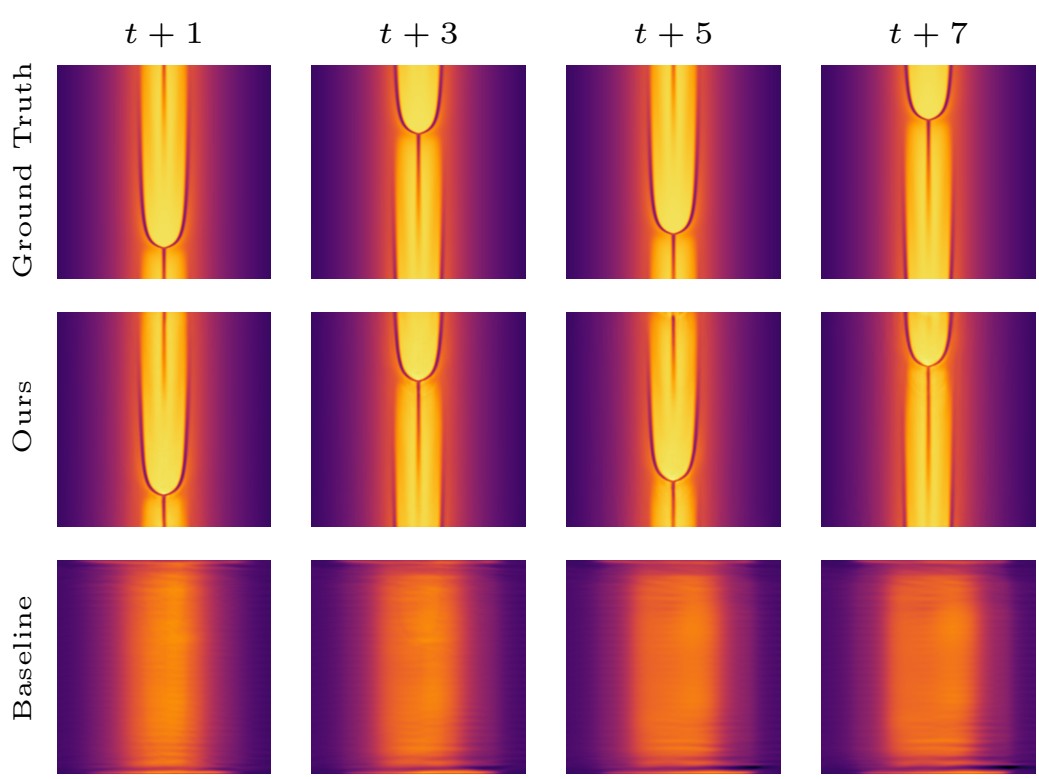

Figure 6: **Qualitative Comparisons on `viscoelastic_instability` Dataset.** We compare the prediction of PhysiX with the ground truth and the prediction of the best baseline model at lead times of 1,3,5,7 frames.

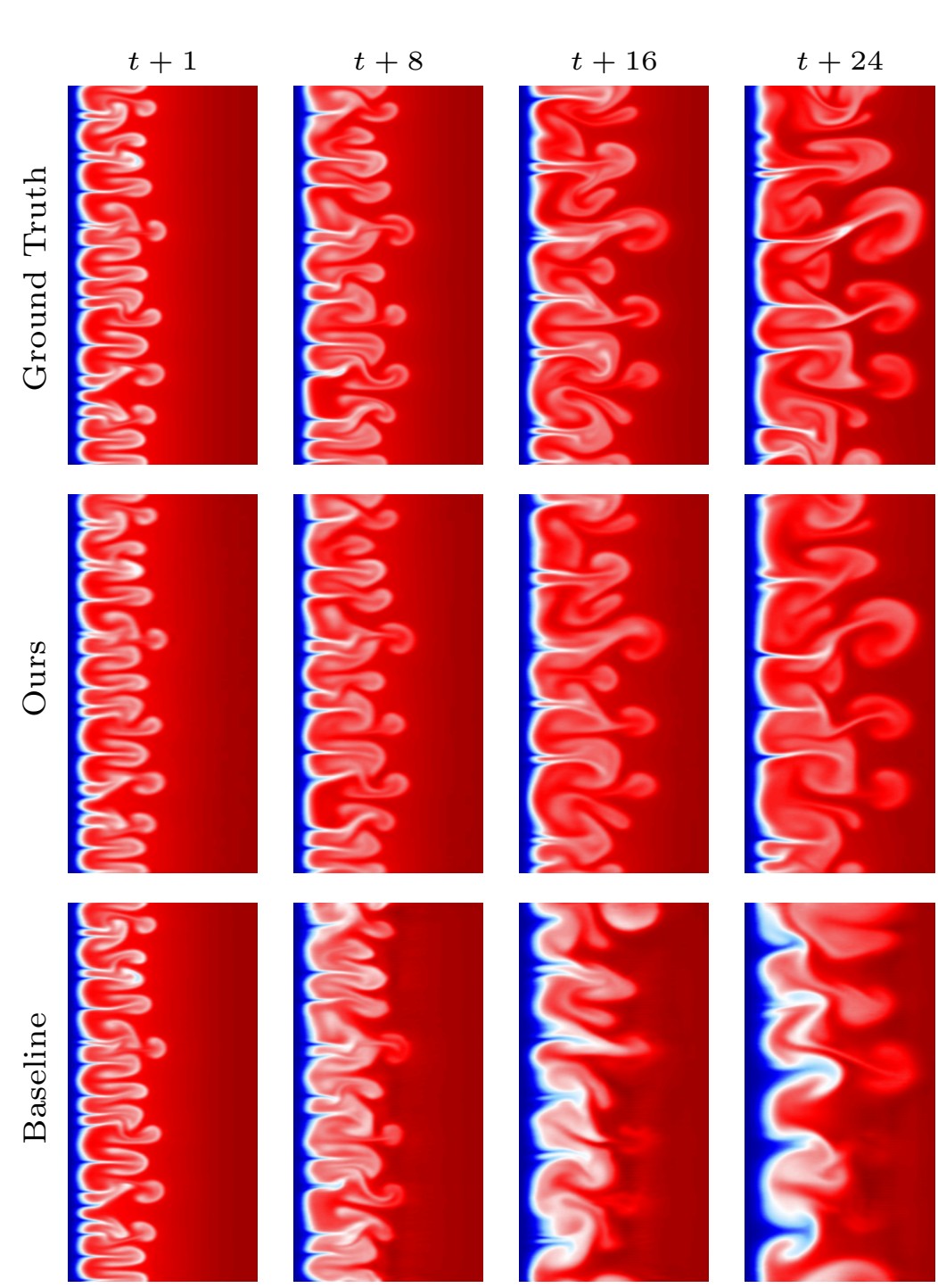

Figure 7: **Qualitative Comparisons on `rayleigh_benard` Dataset.** We compare the prediction of PhysiX with the ground truth and the prediction of the best baseline model at lead times of 1,8,16,24 frames.

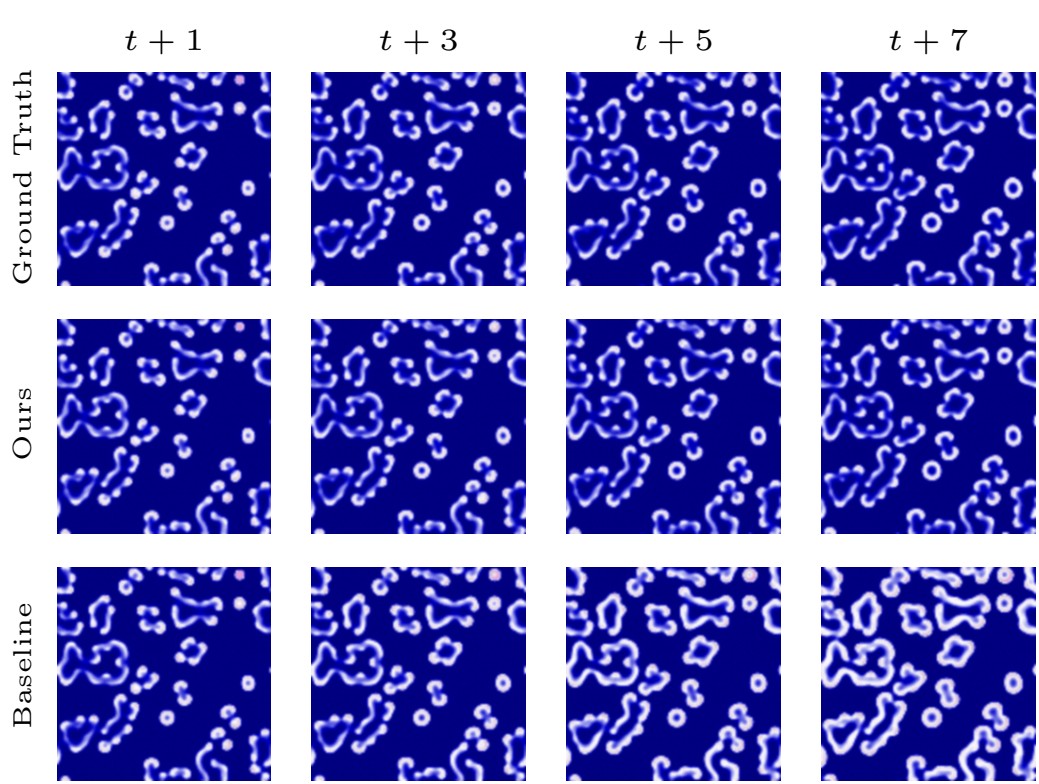

Figure 8: **Qualitative Comparisons on `gray_scott_reaction_diffusion` Dataset.** We compare the prediction of PhysiX with the ground truth and the prediction of the best baseline model at lead times of 1,3,5,7 frames.

