# OpenReview forum: "PhysiX: A Foundation Model for Physics Simulations"
_ICLR.cc/2026/Conference — Submitted to ICLR 2026_

### Official Review · Reviewer_Xbsc · 2025-10-30

**Soundness:** 3
**Presentation:** 3
**Contribution:** 2
**Rating:** 4
**Confidence:** 4

**Summary:**

PhysiX is a family of autoregressive transformer models for physics simulations on 2D regular grids. A tokenizer converts data to a sequence of tokens, which are quantized. An autoregressive causal transformer performs next-token prediction and tokens are mapped back to the phyiscs space via a decoder. A refinement module corrects discretization errors. Comprehensive experiments on The Well are performed.

**Strengths:**

- The paper is well written and follows a clear structure.
- Strong performance on benchmarks from The Well.
- Interesting experiments and adequate ablation studies.

**Weaknesses:**

- The paper does not compare to, discuss or mention previous approaches that have used causal autoregressive transformers for 2D physics simulations, such as [1,2,3].

- Experiments and results are strong, but the novelity of the approach itself is incremental at best. I would appreciate a deeper discussion on the modifications/differences to [1,2,3], ideally also quantitatively; but I also acknowledge that this might be out of the scope of the rebuttal.

- I see the inclusion of the refinement module as a disadvantage rather than an advantage. If I understand correctly, once PhysiX is trained without the refinement module, a new dataset is generated based on the predictions of the trained PhysiX network on which the refinement module is trained. That is a lot of extra work and it's also something that could be done for all baselines. Therefore, for a fair comparison, I would either not include the refinement module for the comparison, or do the same for all the baselines.

- Could you include an architecture benchmark table comparing GFLOPs/VRAM/throughput for PhysiX and all baselines?

- U-Net and C-U-Net from The Well do not represent sota U-Net architectures. I expect that using any modern U-Net for pixel space diffusion gives improved performance, e.g. [4] I am also surprised by the bad performance of Poseidon on the benchmarks, are you sure the network had converged? What were your training hyperparameters?

- The title is overselling it to me; I think limitations such as "2D" should be included in the title


[1] https://arxiv.org/abs/2501.18972

[2] https://arxiv.org/abs/2406.04501

[3] https://arxiv.org/pdf/2410.03437

[4] https://arxiv.org/abs/2301.11093

**Questions:**

- What is the advantage of the quantization of tokens for PDEs/physics simulations? I would understand that the quantization could be useful for training a probabilistic model, but PhysiX is not trained on data that would require this; in addition to that, it slows down inference, since inference has to generate each token after another, not allowing to generate multiple tokens in parallel, which is e.g. done in [1].
- How did you deal with the normalization of the dataset? The difference in scale, even within the same dataset of The Well can be very large. Did you use any weighting in the loss (not the stratified sampling strategy). Could you expand the training details in the appendix; adding some details like specific loss functions for tokenizer/autoregressive transformer?
- The unified framework for the input channels makes sense; and works well when all downstream tasks have the same types of fields/channels. As you already mentioned in the paper, this makes zero-shot/finetuning more difficult, since the tokenizer always has to be finetuned. Another disadvantage is that the initial embedding can become very large. For example, the dataset active matter has 10+ fields. How big is the first embedding layer/union of all channels in practice for TheWell?
- In line 354, "MPP and DPOT interleave space and time modeling", can you please clarify this?

---

> ### Author Response · Authors · 2025-11-25
>
> We thank Reviewer Xbsc for their thoughtful review. We are encouraged that you found the paper well-written and structured, and we appreciate your recognition of the strong performance on The Well benchmark as well as the adequacy of our ablation studies. We address your specific concerns and questions below.
>
> ### Weaknesses
>
> > Discussion on the modifications/differences of PhysiX to [1,2,3]
>
> We thank the reviewer for pointing out these relevant recent works. We have added a detailed discussion in the related works section comparing PhysiX to these approaches. To summarize our position:
> - PhysiX vs. BCAT [1]: While both methods use autoregressive transformers, there are three key differences: 1) Latent vs. Pixel: BCAT operates in the original pixel domain, whereas PhysiX operates in a discrete latent space. This allows PhysiX to be more computationally efficient when scaling to high-resolution simulations; 2) Heterogeneity Strategy: BCAT uses zero-padding to match the task with the maximum channel count. PhysiX uses a "union of channels" approach with learnable embeddings. This ensures distinct physical quantities (e.g., pressure vs. concentration) are handled via dedicated embeddings rather than sharing weights, avoiding semantic confusion; and 3) Causality: BCAT uses block-causal masking, whereas PhysiX is purely causal. While block-causal is efficient, purely causal modeling makes PhysiX compatible with pretrained video foundation models (like Cosmos), enabling the transfer learning capabilities shown in Section 4.4.
> - PhysiX vs. Zebra [3]: Zebra also uses discrete tokenization but focuses on in-context learning (ICL) to adapt to varying coefficients, boundary conditions, and forcing terms within the same physical system. To do this, Zebra includes multiple trajectories of the same physical system and trains the model to perform in-context learning prediction. However, Zebra did not discuss how to train the model on diverse physical systems, e.g., how to handle input data heterogeneity. PhysiX, in contrast, focuses on generalization across entirely distinct physical simulations (e.g., transferring from fluid dynamics to reaction-diffusion).
> - PhysiX vs. Fluid-LLM [2]: Unlike PhysiX and the above methods, Fluid-LLM is not autoregressive. It leverages the backbone of a pretrained LLM to output the hidden state of the next prediction step and decodes it using a graph-based decoder. It does not learn to predict multiple future frames.
>
> > Discussion on the importance of the refinement module
>
> Your understanding of the refinement module is correct. We believe the necessity of this module is unique to latent modeling methods such as PhysiX, since the reconstruction error is an inherent bottleneck of tokenization, regardless of the transformer model’s predictive ability.
>
> We also note that even without the refinement module, PhysiX remains highly competitive. As shown in the table below, the refinement module was only critical for 2 out of 8 tasks (gray_scott_reaction_diffusion and turbulent_radiative_layer_2D) where high-frequency artifacts were prominent (see https://imgur.com/a/iTiKoh3 for visualization). Even without refinement, PhysiX outperforms the best baseline (C-U-Net) on 5/8 tasks, with significant margins in shear_flow and rayleigh_benard.
>
> | Task                          | PhysiX w/ refinement | PhysiX w/o refinement | C-U-Net  |
> |-------------------------------|----------------------|------------------------|----------|
> | shear_flow                    | 0.0700               | 0.0713                 | 0.8080   |
> | rayleigh_benard               | 0.1470               | 0.1477                 | 0.6699   |
> | acoustic_scattering (maze)    | 0.0960               | 0.0984                 | 0.0153   |
> | turbulent_radiative_layer_2D  | 0.2098               | 0.3430                 | 0.1956   |
> | active_matter                 | 0.0904               | 0.0913                 | 0.1034   |
> | gray_scott_reaction_diffusion | 0.0210               | 0.2290                 | 0.1761   |
> | viscoelastic_instability      | 0.2370               | 0.2397                 | 0.2499   |
> | helmholtz_staircase           | 0.0180               | 0.0181                 | 0.02758  |

---

> ### Author Response · Authors · 2025-11-25
>
> > Inference efficiency of PhysiX and the baselines
>
> The table below shows the inference efficiency of PhysiX and the baseline methods, evaluated on one-step prediction for the active_matter task.
>
> | Method   | Speed (frames/s) | Peak memory usage (GBs) | GFLOPs   |
> |---------|------------------|--------------------------|----------|
> | C-U-Net | 96.0             | 0.2                      | 20.75    |
> | MPP     | 9.4              | 2.2                      | 545.57   |
> | DPOT    | 26.0             | 2.3                      | 535.52   |
> | Poseidon| 54.7             | 2.5                      | 143.69   |
> | PhysiX-S| 3.5              | 4.1                      | 818.19   |
> | PhysiX-H| 0.65             | 11.2                     | 15367.39 |
>
> While PhysiX has lower inference throughput due to the autoregressive generation, this is an expected trade-off for the significant performance gains observed (lower VRMSE and better stability). Furthermore, even at 0.65 frames/s of the largest model, PhysiX remains orders of magnitude faster than traditional numerical solvers.
>
> > U-Net and C-U-Net from The Well do not represent sota U-Net architectures
>
> We utilized the specific U-Net and C-U-Net implementations provided by The Well benchmark suite. These baselines were tuned specifically for these physics tasks by the benchmark authors. While newer diffusion-based U-Nets [4] exist, simply swapping the architecture does not guarantee improved performance on PDE tasks without extensive tuning. We believe that by including comparisons against strong, modern PDE-specific baselines like MPP, DPOT, and Poseidon, our evaluation is comprehensive and fair.
>
> > Bad performance of Poseidon on the benchmarks
>
> We used the following training hyperparameters for Posiedon:
> - Model: Poseidon-B
> - Effective batch size: 32
> - Learning rate: 1e-4
> - Learning rate for embedding recovery parameters: 1e-3
> - Learning rate for time embedding parameters: 1e-3
> - Weight decay: 1e-6
> - Scheduler: cosine with warmup ratio of 0.05
> - Trained for 50 epochs
>
> As opposed to PhysiX and other baselines, Poseidon takes only 1 initial condition frame as input and predicts the state after some lead time x, which may be the reason why it underperforms other methods that use multiple historic states. We used the default hyperparameters given by the Poseidon repo, but we doubt the performance could improve significantly with better hyperparameters.
>
> > Limitations such as "2D" should be included in the title
>
> We accept this suggestion and have updated the title and abstract to reflect the focus on 2D simulations. However, we note that the framework itself is not restricted to 2D. The tokenizer can be extended to 3D, or 3D spatial dimensions can be "collapsed" into the channel dimension, i.e., if a simulation has a spatial depth of Z and C channels, we could treat it as a 2D simulation with ZxC channels.
>
> ### Questions
>
> > What is the advantage of the quantization of tokens for PDEs/physics simulations
>
> We see three primary advantages to discrete tokenization in this domain:
> - Compatibility: It aligns the physics model with the architecture of state-of-the-art video and language foundation models. This compatibility is precisely what allowed us to leverage the pretrained weights of the Cosmos video model to significantly boost performance (Section 4.4).
> - Uncertainty Quantification: Quantization facilitates probabilistic modeling, which is crucial for scientific domains where data is scarce, leading to imperfect model predictions.
> - Ensembling: It allows for sampling-based generation. In early experiments, we found that generating multiple rollouts and averaging them consistently improved VRMSE (though we reported single-trajectory results for fair comparison with deterministic baselines).
>
> > Data normalization and loss function
>
> We have updated the paper with the following details:
> - Normalization: For each input channel in each dataset, we computed the global mean and standard deviation across the spatial domain and all time steps in the training set. We used these task- and channel-specific statistics to normalize inputs during training and denormalize predictions for evaluation.
> - Loss: We used the standard Mean Squared Error (MSE) loss for both the VAE tokenizer training and the Refinement Module training, without special weighting.

---

> > ### Author Response · Authors · 2025-11-25
> >
> > > Unified framework for the input channels
> >
> > For the 8 datasets in The Well, the union of all unique physical fields results in an initial embedding layer with 46 input channels. This is computationally negligible because it only affects the very first convolution layer, and we found this to work well in practice. If we pretrain on more physical systems and the number of channels grows massively, we can adopt the cross-attention strategy to more effectively learn the interaction between input channels (as previously done in weather forecasting [1, 2]).
> >
> > Regarding fine-tuning: We argue that fine-tuning is structurally inevitable when encountering new physical variables. If a model has never seen a "velocity" field, it cannot zero-shot generalize to it regardless of the embedding strategy.
> >
> > [1] Nguyen, Tung, et al. "ClimaX: A foundation model for weather and climate." International Conference on Machine Learning. PMLR, 2023.
> >
> > [2] Nguyen, Tung, et al. "Scaling transformer neural networks for skillful and reliable medium-range weather forecasting." Advances in Neural Information Processing Systems 37 (2024): 68740-68771.
> >
> > > Clarification of line 354
> >
> > We meant that these architectures decouple spatial and temporal processing:
> > - MPP uses decomposed attention, where layers alternate between space-only or time-only attention.
> > - DPOT typically uses a temporal aggregation layer followed by spatial Fourier layers.
> >
> > While efficient, these approaches prevent the model from learning simultaneous spatiotemporal relationships. PhysiX's unified attention attends to the joint spatiotemporal context, which enhances expressivity for complex dynamics.

---

> > > ### Author Response · Authors · 2025-12-01
> > > **Reminder of our rebuttal**
> > >
> > > Dear Reviewer Xbsc,
> > >
> > > We are writing to follow up on our rebuttal, in which we have done our best to address your concerns with additional experiments and detailed clarifications. Specifically, we have:
> > > - Included a discussion on the modifications/differences of PhysiX to related works suggested by the reviewer.
> > > - Discussed the importance of the refinement module.
> > > - Reported the inference efficiency of PhysiX compared to other methods.
> > > - Clarified the performance of the U-Net and Poseidon baselines.
> > > - Discussed the benefits of generative modeling for PDEs.
> > > - Clarified data normalization, loss function, and the unified framework for handling input channels.
> > >
> > > We sincerely hope you have a moment to review our response. We are eager to address any remaining questions you might have and would be happy to discuss further. Thank you again for your valuable feedback.
> > >
> > > Best regards,
> > >
> > > The Authors

---

### Official Review · Reviewer_mxCq · 2025-10-30

**Soundness:** 4
**Presentation:** 3
**Contribution:** 3
**Rating:** 8
**Confidence:** 3

**Summary:**

This paper introduces a foundation model for physics simulations, targeting a model that would be applicable to a relatively broad set of common PDE families. The overall approach is familiar, with transformer and CNN blocks. The proposed approach provides a few targeted improvements: better tokenization across multiple tasks and variables, a refinement layer for denoising, and transfer learning from non-science video models. The results are presented on the Well benchmark and generally compare very favorably to previous methods.

**Strengths:**

The experiments are fairly complete, considering next-frame predictions, long-timeframe predictions, single and multi-task models, the effect of refinement, scaling across different model sizes, and generalization on unseen tasks.

The results on the Well are quite strong, improving not only against previous approaches, but continuing to show stronger evidence of generalizability across tasks/PDEs.

**Weaknesses:**

At times, the discussion of Refinement Module makes it seem like a super-resolution task, going from a coarse prediction to a higher-resolution one? And it seems like this is true at least in terms of precision but Figure 2 doesn’t give me the impression that it is predicting a longer token sequence than the autoregressive model did.

Compared with the decomposed attention in MPP, other studies have previously suggested advantages to using a unified autoregressive transformer versus decomposing it. To what extent does this change contribute to the improvement of PhysiX versus other approaches?

Relatedly, despite section 4.3, the paper still feels like it needs an ablation test. I don’t have a good feel at the end of how important, if at all, stuff like the new tokenization and embedding approaches were.

**Questions:**

How much does the additional compute required for the refinement module compare to using smaller patches in the first place?

To what extent is the approach expected to change, if at all, for 4D (3D spatial + 1D time) data?

---

> ### Author Response · Authors · 2025-11-25
>
> We thank Reviewer mxCq for their positive assessment of our work. We appreciate that you found our experiments complete and recognized the strength of our results on The Well benchmark, particularly regarding the evidence of generalizability across diverse physical tasks. We address your specific concerns and questions below.
>
> ### Weaknesses
>
> > Clarification on the refinement module
>
> We clarify that the refinement module does not perform super-resolution (upscaling). Rather, it performs a denoising/correction task at the original resolution. The discrete VAE tokenizer introduces quantization noise, such as distinct "blocky" artifacts, that can degrade downstream simulation precision.
>
> We view the refinement module as a lightweight post-processing step to correct these reconstruction errors. Empirically, we found this was not necessary for all tasks; it provided significant benefits in only 2 of the 8 datasets (turbulent_radiative_layer_2D and gray_scott_reaction_diffusion), where the VAE tokenizer introduced specific high-frequency artifacts (see https://imgur.com/a/iTiKoh3 for gray_scott_reaction_diffusion). For the other tasks, the base performance of the autoregressive model was already sufficient.
>
> > Other studies have previously suggested advantages to using a unified autoregressive transformer versus decomposing it. To what extent does this change contribute to the improvement of PhysiX versus other approaches?
>
> We agree with the premise of this question. While decomposed attention (as used in MPP) reduces memory usage and improves computational throughput by separating spatial and temporal mixing, it inherently restricts the model's expressivity.
>
> In complex physics simulations, spatial and temporal dynamics are often inextricably coupled (e.g., advection in fluid dynamics). A unified architecture allows the model to learn relationships across the full spatiotemporal context simultaneously. We attribute a portion of PhysiX's performance gain over MPP (shown in Table 1) to this architectural choice, as it allows the model to capture these intricate spatiotemporal dependencies that decomposed attention might miss.
>
> > Importance of the new tokenization and embedding
>
> Many of our design choices of the tokenizer arise naturally from first principles, such as the 3D temporally causal convolutions for spatiotemporal data, and learnable paddings to handle heterogeneity in the input data across different datasets. We did not explore alternative solutions since these designs were simple and worked well in practice.
>
> ### Questions
>
> > How much does the additional compute required for the refinement module compare to using smaller patches in the first place?
>
> Computationally, we believe training an additional refinement module is a much easier task. For example, halving the spatial patch dimensions increases the sequence length of the transformer model by 4. Since the self-attention mechanism in Transformers scales quadratically, this would result in a $16\times$ increase in attention cost and significantly higher memory usage.
>
> > To what extent is the approach expected to change, for 4D (3D spatial + 1D time) data?
>
> The autoregressive Transformer component would effectively remain unchanged, as it operates on a sequence of tokens regardless of their source dimensionality. The primary change would occur in the Tokenizer:
> - Tokenizer Architecture: We would need to extend the encoder/decoder to handle 3D spatial volumes (using 4D convolutions: 3D space + 1D time).
> - Simplified Adaptation: A more immediate approach to adapting PhysiX to 3D spatial data would be to "collapse" the third spatial dimension into the channel dimension. For example, if a simulation has a spatial depth of Z and C channels, we could treat it as a 2D simulation with ZxC channels. Our "union of channels" embedding strategy could naturally handle this increased channel count without requiring architectural changes to the Transformer.

---

### Official Review · Reviewer_vMxC · 2025-10-31

**Soundness:** 2
**Presentation:** 2
**Contribution:** 2
**Rating:** 4
**Confidence:** 4

**Summary:**

This paper explores a foundation-model approach to simulating physical phenomena across domains. It introduces a universal tokenizer to unify heterogeneous tasks, followed by an autoregressive Transformer and a refinement module after decoding. Experiments indicate consistent gains over baselines in most settings.

**Strengths:**

* The proposed method handles diverse simulation tasks in a unified framework.
* It outperforms the baseline methods in most cases.

**Weaknesses:**

1. Missing long-horizon visualized results. The supplementary material visualizes at most 24 rollout steps, whereas Table 3 reports predictions up to 56 frames. Please provide additional qualitative results for long-term predictions (e.g., full-trajectory videos or densely sampled frames) to assess temporal faithfulness and smoothness. Quantitative metrics for long-horizon predictions alone are insufficient to evaluate dynamic fidelity.
2. Clarify “long-horizon” scope and report full-length rollouts. Table 3 shows up to 56 frames. Is 56 frames the maximum sequence length in the dataset? If not, please include full-length rollout results (per-sequence) to substantiate long-term stability and error accumulation claims. An example from
3. Inference efficiency. Please report inference speed and memory usage relative to baselines, ideally across multiple sequence lengths.
4. Relation to TIE [1]. Since TIE is a Transformer-based neural simulator for physics simulation, including a discussion of the relations to TIE would benefit this paper.


[1]. Shao, et al. Transformer with Implicit Edges for Particle-based Physics Simulation. ECCV 2022.

**Questions:**

The approach appears to be a direct transfer of video-prediction techniques to physics simulation. While data imbalance is a concern in both simulation and video generation, please clarify which simulation-specific adaptations were introduced and how they differ from the video generation.

---

> ### Author Response · Authors · 2025-11-25
>
> We thank the reviewer for their constructive feedback and for recognizing the strong performance of our proposed method. We address your specific concerns and questions below.
>
> > Missing long-horizon visualized results
>
> We have uploaded video visualizations for the 8 tasks considered in our paper at https://drive.google.com/drive/folders/1J_qYkGB6MYVUVPAxnFMsyMunLIVA3vTW?usp=sharing. PhysiX achieves much higher-fidelity, more accurate, and more stable predictions than the baselines across all tasks and samples.
>
> > Clarify “long-horizon” scope and report full-length rollouts
>
> Different tasks in The Well have different numbers of time steps. Among the tasks we considered in this paper, only gray_scott_reaction_diffusion (1000), rayleigh_benard (200), and shear_flow (200) have more than 100 steps. Due to the limited time of the rebuttal, we reported the performance of PhysiX and the best baseline up to 100 steps ahead in the table below. PhysiX significantly outperforms the baseline in 5/6 tasks, demonstrating the strong stability of our proposed method.
>
> | Dataset                        | t=57:76 PhysiX | t=57:76 Baseline | t=77:95 PhysiX | t=77:95 Baseline |
> |--------------------------------|----------------|------------------|----------------|------------------|
> | shear_flow                     | **0.455**      | 20.482           | **0.463**      | 20.482           |
> | rayleigh_benard                | **0.897**      | 1.236            | **0.963**      | 1.017            |
> | acoustic_scattering (maze)     | 2.798          | **1.774**        | 3.001          | **2.750**        |
> | active_matter                  | **1.309**      | 1.665            | -              | -                |
> | turbulent_radiative_layer_2D   | **1.073**      | 1.379            | **0.988**      | 1.340            |
> | gray_scott_reaction_diffusion  | **0.964**      | 23.072           | **1.341**      | 32.048           |
>
>
> > Inference efficiency
>
> The table below shows the inference efficiency of PhysiX and the baseline methods, evaluated on one-step prediction for the active_matter task.
>
> | Method   | Speed (frames/s) | Peak memory usage (GBs) | GFLOPs   |
> |---------|------------------|--------------------------|----------|
> | C-U-Net | 96.0             | 0.2                      | 20.75    |
> | MPP     | 9.4              | 2.2                      | 545.57   |
> | DPOT    | 26.0             | 2.3                      | 535.52   |
> | Poseidon| 54.7             | 2.5                      | 143.69   |
> | PhysiX-S| 3.5              | 4.1                      | 818.19   |
> | PhysiX-H| 0.65             | 11.2                     | 15367.39 |
>
>
> While PhysiX has lower inference throughput due to the autoregressive generation, this is an expected trade-off for the significant performance gains observed (lower VRMSE and better stability). Furthermore, even at 0.65 frames/s of the largest model, PhysiX remains orders of magnitude faster than traditional numerical solvers.
>
> > Including a discussion of the relations to TIE would benefit this paper.
>
> We thank the reviewer for this suggestion and have added a discussion of TIE to the paper. While we share the high-level motivation of applying transformers to simulation, there are three fundamental differences between the approaches:
> - Application Domain: TIE focuses on particle-based dynamics, primarily for computer graphics applications. In contrast, PhysiX focuses on grid-based PDE solvers for scientific physical systems.
> - Architectural Philosophy: TIE proposes a specialized architecture tailored to capture complex particle interactions. PhysiX intentionally leverages a standard autoregressive transformer architecture on a discrete latent space. This design choice is critical as it ensures compatibility with existing large-scale video foundation models, enabling the transfer learning capabilities we demonstrate in Section 4.4.
> - Generalization Scope: TIE focuses on learning dynamics within specific systems. It does not address the challenge of generalization across multiple, heterogeneous simulation tasks, which is the primary contribution of our work.

---

> > ### Author Response · Authors · 2025-12-01
> > **Reminder of our rebuttal**
> >
> > Dear Reviewer vMxC,
> >
> > We are writing to follow up on our rebuttal, in which we have done our best to address your concerns with additional experiments and detailed clarifications. Specifically, we have:
> > - Conducted additional quantitative evaluation and qualitative visualizations of PhysiX on long-horizon predictions.
> > - Compared the inference efficiency of PhysiX with other methods.
> > - Included a discussion on the relation to TIE in the paper.
> >
> > We sincerely hope you have a moment to review our response. We are eager to address any remaining questions you might have and would be happy to discuss further. Thank you again for your valuable feedback.
> >
> > Best regards,
> >
> > The Authors

---

### Official Review · Reviewer_dRQY · 2025-10-31

**Soundness:** 2
**Presentation:** 2
**Contribution:** 2
**Rating:** 2
**Confidence:** 4

**Summary:**

The paper introduces PhysiX, a family of foundation models designed for physics simulations. The architecture comprises three key components: a U-Net–style tokenizer, an autoregressive transformer, and a ConvNeXt-U-Net refinement module.
These components are trained sequentially. First, the tokenizer is trained on all datasets to map spatiotemporal solutions into a universal set of discrete tokens. Next, the autoregressive transformer learns to make next-token predictions. Finally, the refinement module is trained to correct quantization errors from tokenization and reconstruct fine-scale physical details.
PhysiX models are trained and evaluated on spatiotemporal solutions of 2D physical systems from The Well benchmark, and compared against results reported in the original Well paper. Across most tasks, PhysiX achieves lower VRMSE, demonstrating improved accuracy.

**Strengths:**

1. The work addresses an important research topic. Developing a foundation model that generalizes across physical systems would have great impact on the sciML community.
2. The technical designs are reasonable, for example, temporal causality is enforced through causal padding.

**Weaknesses:**

1. The primary concern is that the contribution of the work appears limited. Algorithmically, it combines established components rather than introducing fundamentally novel ideas. In terms of pretraining generalizability, the model is trained on only eight 2D physical systems from The Well. Similar efforts, such as MPP and Poseidon, already exist.
2. Key model architecture hyperparameters are not reported, which makes it hard for reproducibility. Details regarding the training and testing procedures are also missing.
3. The improvements on unseen simulations in Section 4.5 are minimal, making it difficult to convincingly demonstrate the benefits of pretraining.
4. It's unclear how the next frame is generated. Is the model making multiple autoregressive next-token predictions per frame, given that each frame maps to multiple tokens? If so, how do the training and inference costs compare to those of MPP?

**Questions:**

1. What is the motivation for training a separate ConvNeXt-U-Net refinement module for each dataset? Would a single shared refinement module be sufficient? Additionally, have you evaluated the relative contributions of the tokenizer and autoregressive transformer VS the refinement module, given that the ConvNeXt-U-Net in the original paper performed well on several tasks already?
2. Did the use of causal padding in temporal convolutions affect prediction accuracy? If so, what differences were observed?
3. What were the exact training and evaluation setups? Specifically, was the model trained purely for next-token prediction, or in an autoregressive manner?
4. Did the 4B-parameter models exhibit signs of overfitting?

---

> ### Author Response · Authors · 2025-11-25
>
> We thank the reviewer for their constructive feedback and for recognizing the significance of our work in developing foundation models for scientific machine learning. We are glad you appreciated our technical design choices, such as the enforcement of temporal causality, and acknowledged the potential impact of a model that generalizes across physical systems. We address your specific concerns and questions below.
>
> ### Weaknesses
>
> > The primary concern is that the contribution of the work appears limited. Algorithmically, it combines established components rather than introducing fundamentally novel ideas.
>
> We respectfully disagree with the assessment that the work lacks novelty because it combines established components. Per the ICLR review guidelines, novelty includes "creative combinations of existing ideas" and "application to a new domain." The originality of PhysiX lies in the non-trivial combination of discrete tokenization and autoregressive transformers – techniques proven in vision/language – to solve specific bottlenecks in physics simulations (specifically data scarcity and heterogeneity).
>
> Moreover, this application requires significant adaptation. For example:
> - Data heterogeneity: Unlike natural images, physics datasets have varying channel dimensions and physical meanings. PhysiX introduces a "union of channels" embedding strategy (Section 3.1) to train jointly on diverse systems without manual alignment.
> - High precision: Physics simulations are intolerant of quantization noise that would be acceptable in video generation. We introduce a refinement strategy specifically to address the reconstruction errors inherent to discrete tokenization in dynamical systems.
>
> These are not "out-of-the-box" applications of existing algorithms but rather a targeted architectural adaptation that enables a successful transfer of foundation models in language/vision to physics simulation.
>
> > The model is trained on only eight 2D physical systems from The Well. Similar efforts, such as MPP and Poseidon, already exist.
>
> While we acknowledge the existence of concurrent works, we emphasize two points regarding scale and performance:
> - Scale and Diversity: Training on these 8 tasks is a significant undertaking. Collectively, they constitute 1.7TB of data, covering a highly diverse set of physical phenomena ranging from fluid dynamics (Navier-Stokes) to chemical reaction-diffusion systems. This diversity presents a rigorous test for a unified model.
> - Performance Matters: While PhysiX, MPP, and Poseidon share the high-level goal of multi-physics learning, the execution and results differ. As shown in Table 1 and Table 2, PhysiX significantly outperforms these baselines (SOTA on 5/8 datasets for next-frame prediction and superior long-horizon stability). Just as LLM research continues to push performance despite shared underlying architectures, we believe demonstrating a general-purpose AR model can outperform specialized physics transformers is a critical contribution to the community.
>
> > Key model architecture hyperparameters are not reported
>
> We apologize for the omission. We have updated the paper (Appendix B) with the detailed architectural hyperparameters for PhysiX. We are also committed to releasing our code and model checkpoints upon acceptance to ensure full reproducibility.
>
> > The improvements on unseen simulations in Section 4.5 are minimal
>
> We realized that our initial comparison in Table 5 was unfair to the "scratch" baseline. In the original submission, both PhysiX_f (finetuned) and PhysiX_s (scratch) used the same VAE tokenizer (which was pretrained). This gave the scratch model an unfair advantage by leveraging a pretrained component.
>
> To rectify this, we performed a rigorous re-evaluation where we trained new VAE tokenizers from scratch for the unseen tasks (euler_multi_quadrants and acoustic_scattering) and then trained the transformer on top. The results below demonstrate that under this fairer comparison, PhysiX_f significantly outperforms the scratch baseline, confirming the strong benefit of pretraining. We added this experiment to the paper.
>
> | Models   | euler_multi_quadrants (periodic b.c.) |   |   |   | acoustic_scattering (discontinuous) |   |   |   |
> |----------|----------------------------------------|---|---|---|--------------------------------------|---|---|---|
> |          | t=1   | t=2:8 | t=9:26 | t=27:56 | t=1   | t=2:8 | t=9:26 | t=27:56 |
> | PhysiX_f | **0.105** | **0.188** | **0.358**  | **0.642**   | **0.038** | **0.057** | **0.443**  | **1.168**   |
> | PhysiX_s | 0.205 | 0.303 | 0.464  | 0.754   | 0.078 | 0.114 | 0.689  | 1.554   |

---

> > ### Author Response · Authors · 2025-11-25
> >
> > > It's unclear how the next frame is generated
> >
> > PhysiX generates predictions token-by-token, analogous to an LLM. During training, the model sees a sequence of 13 frames, which the tokenizer compresses into a sequence of 5xH/8xW/8 latent tokens. The tokens corresponding to the first 5 frames serve as context.
> >
> > During inference, we provide the model with the latent tokens of 5 context frames. The model then autoregressively generates the tokens corresponding to the next 8 frames. To extend the simulation further (long-horizon rollout), we simply use the last 5 generated frames as the new input context and repeat the process. We have added this clarification to Section 3.2.
> >
> > > How do the training and inference costs compare to those of MPP?
> >
> > Training: PhysiX is competitive with MPP in training time.
> > - PhysiX-S (250M): ~3 hours (8x A100, 10k iterations).
> > - PhysiX-H (4B): ~19.5 hours (8x A100, 10k iterations).
> > - MPP: ~14 hours (8x A100, 10k iterations).
> >
> > Inference (measured on active_matter):
> > - PhysiX-S: 3.5 frames/s (4GB VRAM).
> > - PhysiX-H: 0.65 frames/s (11GB VRAM).
> > - MPP: 9 frames/s (2GB VRAM).
> >
> > While PhysiX has lower inference throughput due to the autoregressive generation, this is an expected trade-off for the significant performance gains observed (lower VRMSE and better stability). Furthermore, even at 0.65 frames/s of the largest model, PhysiX remains orders of magnitude faster than traditional numerical solvers.
> >
> > ### Questions
> >
> > > What is the motivation for training a separate ConvNeXt-U-Net refinement module for each dataset? Would a single shared refinement module be sufficient?
> >
> > It is not a strict requirement to train separate modules. We treated refinement as a lightweight, dataset-specific post-processing step to remove tokenizer artifacts. In our experiments, we found that the refinement module was only necessary for 2 out of the 8 tasks (turbulent_radiative_layer_2D and gray_scott_reaction_diffusion), where the VAE tokenizer introduced significant artifacts that degraded performance. For the other tasks, the base performance without refinement was already sufficient. In practice, one could train a single universal refiner, but dataset-specific training was more computationally efficient given that only specific tasks required it.
> >
> > > Contributions of the tokenizer and autoregressive transformer VS the refinement module
> >
> > Please see below for the comparison of PhysiX with and without the refinement module. We observed that the refinement module primarily contributes to performance on the two tasks mentioned above. For example, in gray_scott_reaction_diffusion, the tokenizer introduced "blocky" artifacts that degraded VRMSE, and the refinement module effectively smoothed these out (see https://imgur.com/a/iTiKoh3). For tasks without significant high-frequency tokenizer noise, the autoregressive transformer carries the bulk of the predictive errors.
> >
> > | Task                           | PhysiX w/ refinement | PhysiX w/o refinement |
> > |--------------------------------|----------------------|------------------------|
> > | shear_flow                     | 0.0700               | 0.0713                 |
> > | rayleigh_benard               | 0.1470               | 0.1477                 |
> > | acoustic_scattering (maze)     | 0.0960               | 0.0984                 |
> > | turbulent_radiative_layer_2D   | 0.2098               | 0.3430                 |
> > | active_matter                  | 0.0904               | 0.0913                 |
> > | gray_scott_reaction_diffusion  | 0.0210               | 0.2290                 |
> > | viscoelastic_instability       | 0.2370               | 0.2397                 |
> > | helmholtz_staircase            | 0.0180               | 0.0181                 |
> >
> >
> > > Did the use of causal padding in temporal convolutions affect prediction accuracy? If so, what differences were observed?
> >
> > Since our architecture relies on an autoregressive transformer in the latent space, temporal causality in the tokenizer is a structural requirement, not an optional hyperparameter. We did not experiment with removing it because doing so would violate the causality required for the next-token prediction objective.
> >
> > > What were the exact training and evaluation setups? Specifically, was the model trained purely for next-token prediction, or in an autoregressive manner?
> >
> > The model was trained purely using a next-token prediction objective. Given a sequence of 13 frames 13xHxW, the VAE tokenizer compresses this to a latent sequence of length 5xH/8xW/8. We use the tokens corresponding to the first 5 frames as input context (no loss computed) and train the model to predict the tokens for the subsequent 8 frames. We added this detail to Section 3.2 of the paper.

---

> > > ### Author Response · Authors · 2025-11-25
> > >
> > > > Did the 4B-parameter models exhibit signs of overfitting?
> > >
> > > We observed overfitting only when training the 4B model from scratch. However, when initializing the 4B model from the pre-trained video generation model (Cosmos), we observed consistent decreases in validation loss without signs of overfitting. This confirms our hypothesis in Section 4.4 that transfer learning from natural video is the key enabler for scaling physics models to this size.

---

> > > > ### Author Response · Authors · 2025-12-01
> > > > **Reminder of our rebuttal**
> > > >
> > > > Dear Reviewer dRQY,
> > > >
> > > > We are writing to follow up on our rebuttal, in which we have done our best to address your concerns with additional experiments and detailed clarifications. Specifically, we have:
> > > > - Clarified the novelty and contribution of our work in relation to previous work, like MPP or Poseidon.
> > > > - Conducted additional experiments to highlight the advantage of pretraining.
> > > > - Reported key architecture details and hyperparameters, and training and evaluation setups.
> > > > - Reported the training and inference costs of PhysiX compared to other methods.
> > > > - Clarified the importance of the refinement module.
> > > >
> > > > We sincerely hope you have a moment to review our response. We are eager to address any remaining questions you might have and would be happy to discuss further. Thank you again for your valuable feedback.
> > > >
> > > > Best regards,
> > > >
> > > > The Authors

---

### Meta-Review · Area_Chair_49xT · 2026-01-04

**Summary:**

This paper receives 1 positive rating and 3 negative ratings. The main concerns lie in 1) the limited novelty, 2) the motivation and necessity of task-specific refinement module, 3) the soundness of the claim "A Foundation Model for Physics Simulations", 4) the long-horizon visualization results.

Authors have provided responses, and no follow-up discussions are available.

After checking the paper, all reviews, and authors' responses, AC feels the scope of evaluated tasks and the temporal duration of evaluated simulations are insufficient to support the foundation model claim, as raised in 3&4 of reviewers' concerns. Authors' responses didn't address this concern. Thus the decision is reject.

**Reviewer Concerns:**

Concern on novelty is addressed partially, but the difference w.r.t. related works is too subtle.

Task-specific refinement module is addressed partially. Although without this module the proposed model can perform well, including this module has somewhat conflicted with the foundation model claim.

Concern on the claim "A Foundation Model for Physics Simulations" has not been addressed.

Concern on long-horizon results is addressed partially, where the duration is still too short for a foundation model.

**Reviewer Scores:**

Reviewer dRQY would change his/her score since most of the concerns are responded properly.

---

### Decision · Program_Chairs · 2026-01-26

Reject